# Self-Healing in Cyber–Physical Systems Using Machine Learning: A Critical Analysis of Theories and Tools

Obinna Johnphill [1], Ali Safaa Sadiq [1,*], Feras Al-Obeidat [2], Haider Al-Khateeb [3], Mohammed Adam Taheir [4], Omprakash Kaiwartya [1,*] and Mohammed Ali [5]

1   Department of Computer Science, Nottingham Trent University, Clifton Lane, Nottingham NG11 8NS, UK; obinna.johnphill2022@my.ntu.ac.uk
2   College of Technological Innovation, Zayed University, Abu Dhabi P.O. Box 144534, United Arab Emirates; feras.al-obeidat@zu.ac.ae
3   Cyber Security Innovation (CSI) Research Centre, Aston Business School, Aston St, Birmingham B4 7ET, UK; h.al-khateeb@aston.ac.uk
4   Faculty of Technology Sciences, Zalingei University, Zalingei P.O. Box 6, Central Darfur, Sudan; mohammedtaher30@yahoo.com
5   Department of Computer Science, King Khalid University, Abha 61421, Saudi Arabia; mabood@kku.edu.sa
*   Correspondence: ali.sadiq@ntu.ac.uk (A.S.S.); omprakash.kaiwartya@ntu.ac.uk (O.K.)

**Abstract:** The rapid advancement of networking, computing, sensing, and control systems has introduced a wide range of cyber threats, including those from new devices deployed during the development of scenarios. With recent advancements in automobiles, medical devices, smart industrial systems, and other technologies, system failures resulting from external attacks or internal process malfunctions are increasingly common. Restoring the system's stable state requires autonomous intervention through the self-healing process to maintain service quality. This paper, therefore, aims to analyse state of the art and identify where self-healing using machine learning can be applied to cyber–physical systems to enhance security and prevent failures within the system. The paper describes three key components of self-healing functionality in computer systems: anomaly detection, fault alert, and fault auto-remediation. The significance of these components is that self-healing functionality cannot be practical without considering all three. Understanding the self-healing theories that form the guiding principles for implementing these functionalities with real-life implications is crucial. There are strong indications that self-healing functionality in the cyber–physical system is an emerging area of research that holds great promise for the future of computing technology. It has the potential to provide seamless self-organising and self-restoration functionality to cyber–physical systems, leading to increased security of systems and improved user experience. For instance, a functional self-healing system implemented on a power grid will react autonomously when a threat or fault occurs, without requiring human intervention to restore power to communities and preserve critical services after power outages or defects. This paper presents the existing vulnerabilities, threats, and challenges and critically analyses the current self-healing theories and methods that use machine learning for cyber–physical systems.

**Keywords:** cyber–physical system; cybersecurity; threat tolerance; self-healing; intrusion detection; machine-learning algorithms





## 1. Introduction

This narrative review paper presents a descriptive review of manuscripts on cyber–physical self-healing systems using machine learning. Cyber–physical systems (CPSs) are integrated systems that bridge the physical and cyber domains, enabling the seamless integration of biological processes and computing systems [1]. Self-healing in CPSs refers to the ability of these systems to automatically detect and respond to faults or failures without human intervention, enhancing their resilience and reliability [2]. While self-healing

capabilities can improve the performance and robustness of CPSs, they also face several vulnerabilities, threats, and challenges that need to be addressed [3]. These include hardware and software component vulnerabilities that may be susceptible to cyber-attacks and other threats, compromising the self-healing process [4]. The lack of standardisation in self-healing mechanisms and technologies creates system interoperability issues, leading to vulnerabilities and integration challenges [2]. The complexity of CPSs, with multiple interconnected components, poses difficulties in identifying and diagnosing faults, making it challenging to implement effective self-healing mechanisms [5]. Human error during system design, implementation, and maintenance can create vulnerabilities and compromise the self-healing capabilities of the system [2]. The lack of visibility into CPS self-healing systems can also hinder fault identification and compromise system operations [6]. CPS self-healing systems are also vulnerable to malicious attacks, including denial-of-service attacks, malware, and hacking, which can compromise the system's integrity and availability [7]. Considering that CPS self-healing systems are vital for critical infrastructure, such as transportation systems and power grids, failures or vulnerabilities in these systems can have severe safety implications [8]. Addressing these vulnerabilities, threats, and challenges is essential to ensure the security, reliability, and safety of critical infrastructure supported by self-healing capabilities in CPSs [2].

The increased adoption of digital systems in conducting human socioeconomic development affairs concerning business, manufacturing, healthcare provisions, and government services comes with the attendant risk of increased threats to computer systems and networks. These threats could be in the form of cyber-attacks on the individual level or at the organisational level. For example, they targeted those isolated at home during the COVID-19 pandemic lockdowns, schools, businesses, hospitals, manufacturing plants, and social infrastructures. Through the widespread adoption of digital systems, communities have become more susceptible to malicious cyber-attacks; hence, the importance of research around computer systems self-healing has increased over the recent years. A review of existing approaches and methodologies has been conducted to address the need for robust self-healing and self-configuring systems to secure cyber–physical systems against security threats. The selection of methods for this review paper was based on a systematic literature search conducted in major scientific databases such as IEEE Xplore, ACM Digital Library, and Google Scholar. Keywords such as "computer systems self-healing", "cyber–physical systems security", and "self-configuring systems" were used to identify relevant articles published in peer-reviewed journals and conference proceedings. A total of 40 articles were included in this review, covering a wide range of topics related to self-healing and self-configuring systems in the context of cyber–physical systems. The selected articles provide insights into the current state-of-the-art challenges and future directions in this field. By examining these approaches, this review aims to contribute to the development of robust self-healing and self-configuring systems for securing current and future cyber–physical systems. Safety within cyber–physical systems cannot be overemphasised as it is not only an economic imperative but, in some cases, such as in a healthcare setting, can become a matter of life or death. There is, therefore, the broad scope for finding solutions that can aid the development of robust self-healing or self-configuring systems capable of securing current and future cyber–physical systems against security threats.

Cyber–physical systems are part of the Industry 4.0 devices that utilise the power of the Internet to convert the existing Industry 3.0 devices into smart industry devices. These include cyber–physical systems deployed in smart manufacturing, smart grid, smart city, and innovative automobiles. The cyber–physical system is highlighted in Figure 1 as part of Industry 4.0, and the figure focuses on the physical components of Industry 4.0, including cyber–physical systems and IoT, while underscoring the self-healing capability of CPSs in modern manufacturing systems using digital technologies such as cloud computing. An example of such development in transitioning the existing state-of-the-art systems protection from manual interventions to a self-healing approach through automation is noted in [9]. The study argues that as providers migrate from 4G to more robust 5G networks,

the operational costs associated with network failures, predicted to increase exponentially, account for approximately 23% to 26% of revenue from the mobile network. A shift towards automating the system's protection process through self-healing is occurring to control expenses as mobile network providers migrate to 5G. Self-healing systems are being deployed in electricity distribution plants worldwide, with most deployments burdened with latency, bandwidth, and scalability problems, as highlighted in [10]. Standardised architecture for distributed power control using self-healing functionality to solve systems faults is presented. The system proposal increases reliability during normal operations and resilience during threat events. The result of the self-healing experiment in [10] is currently undergoing field implementation by Duke Energy. Deploying machine learning to build self-healing functionality into the power grid is very important in a world where population growth is rising, and according to [11], frequent power outages constitute a considerable cost to the economy and adversely affect people's quality of life. A proposal for using a fault-solving library coupled with a machine-learning algorithm to create self-healing functionality in computer systems was put forth by [11].

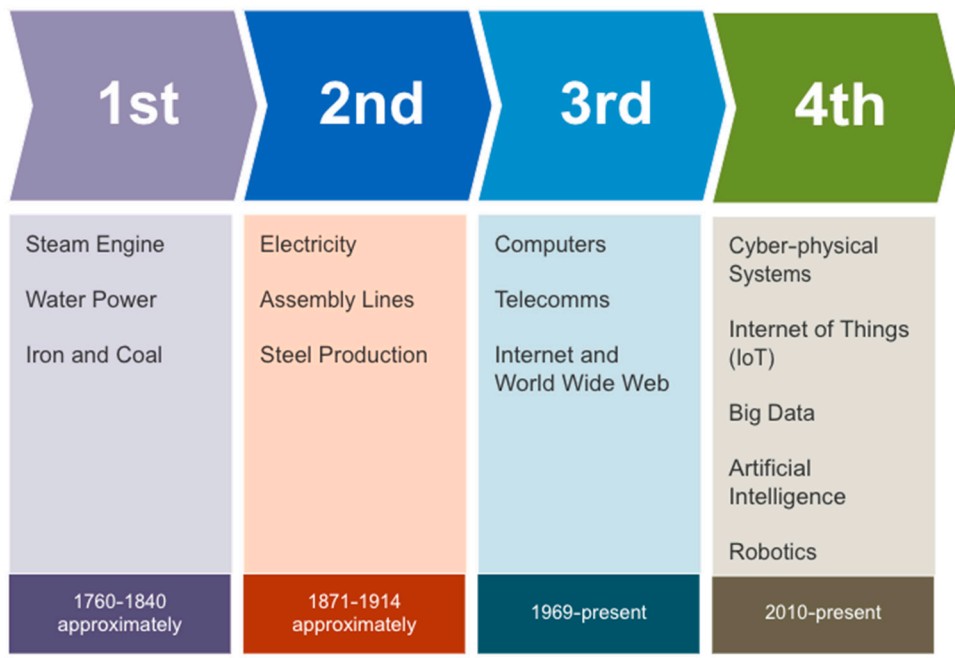

**Figure 1.** Chronological progression of industrial revolutions: from the 1st to the 4th.

This paper briefly discusses the proposal, particularly the twin model approach. Self-healing functionality is vital for providing excellent quality of service (QoS) in cloud computing. With QoS increasingly critical to services offered by vendors of cloud computing, such services as software as service (SaaS), platform as service (PaaS), and infrastructure as service (IaaS), self-healing functions allow the network environment the ability to recover from failure situations that may occur within a software, network, or hardware part of the system, in such cases described in [12]. A technique was developed called self-configuring and self-healing of cloud-based resources RADAR. The principal issue that affects the optimal performance of the smart grid network is multifaceted failures in multiple areas of the network, such as network overload, systems intrusion, systems misconfiguration, etc. These failures can potentially cause severe setbacks to the economy and the quality of human life, which can be mitigated by applying self-healing functionality to the system, as demonstrated in recent research studies. Part of the myriad of solutions that have been proposed is using a fault-solving strategy library on a twin model system and a machine-learning (ML) algorithm to implement a self-healing mechanism in a smart grid. The ML algorithm compiles with the dataset derived from the fault-solving library and is then deployed to detect the anomalies within the cyber–physical system. The anomaly detection

process is the first step towards implementing self-healing functionality and detecting such that the self-healing functionality is triggered after the fault classification process gas is completed and a viable mitigation solution is found within the fault-solving library.

This paper adopts a narrative research method within the qualitative methodology, using existing literature to highlight theories, machine-learning algorithms, and network architectures for implementing self-healing functionality, which can then be deployed to protect the security of cyber–physical systems. The paper surveys existing literature and shows the research areas where similar systems have been implemented and gaps still exist, intending to aid future studies. The goal and objectives of this paper encompass the following four points, which aim to contribute to enhancing knowledge in the field of study:

- This paper aims to enhance knowledge by highlighting current trends in the area of study;
- The main objective of this paper is to identify the latest machine-learning tools, methods, and algorithms for integrating self-healing functionality into cyber–physical systems;
- The self-healing capability of cyber-physical systems will be evaluated concerning state-of-the-art techniques, and machine-learning tools and methods in implementing self-healing functions will be explored;
- The existing literature will be critically reviewed to identify current tools, methods, algorithms, classification models, frameworks, networks, and architectures currently deployed for a self-healing approach.

The implications of the existing approaches for future research are discussed, emphasising how the literature review and findings can contribute to advancing future experiments. This paper's structure includes sections on self-healing theories, self-healing for cyber-physical systems, self-healing methods, an analytical comparison of promising approaches, and a critical discussion of presented theories and techniques. The taxonomy of the literature is summarised in Table 1.

**Table 1.** Taxonomy of the literature.

| Principal Topic | Authors |
|---|---|
| 1. Resilience and Risk Assessment | ■ Cai et al. [11]<br>■ Degeler et al. [13]<br>■ Samir et al. [14]<br>■ Wyers et al. [15]<br>■ Gill et al. [12]<br>■ Mehmet [16] |
| 2. Self-Healing Approaches and Techniques | ■ Chen and Bahsoon [17]<br>■ Singh et al. [18]<br>■ Stojanovic and Stojanovic [19]<br>■ Berry and Chollot [20]<br>■ Schneider et al. [10]<br>■ Khalil et al. [21]<br>■ Hsieh [14]<br>■ El Fallah Seghrouchni et al. [2] |
| 3. Intrusion Detection and Security | ■ Degeler et al. [13]<br>■ Joseph and Mukesh [9]<br>■ Ahmad et al. [22]<br>■ Berry and Chollot [20]<br>■ Zhang et al. [6]<br>■ Subashini and Kavitha [4]<br>■ Colabianchi et al. [23]<br>■ Mohammadi et al. [24] |

**Table 1.** *Cont.*

| Principal Topic | Authors |
|---|---|
| 4. Machine Learning and Artificial Intelligence | ■ Bodrog et al. [2]<br>■ Ali-Tolppa et al. [25]<br>■ Karim et al. [26]<br>■ Ahmad et al. [27]<br>■ Tiwari et al. [28]<br>■ Yang et al. [29]<br>■ Al-juaifari et al. [30]<br>■ Bothe et al. [Bothe] |
| 5. Fault Diagnosis and Detection | ■ Singh et al. [18]<br>■ Li and Li [1]<br>■ Sejdić et al. [5]<br>■ Mohammadi et al. [24] |
| 6. Resilience and Robustness | ■ Idio et al. [31]<br>■ Hahsler et al. [3]<br>■ Breiman [32] |
| 7. Survey and Overview of Cyber-physical Systems | ■ Chen and Bahsoon [17]<br>■ Subashini and Kavitha [4]<br>■ Mahdavinejad et al. [8]<br>■ Samuel and Madria [7]<br>■ Zhang et al. [6] |

## 2. Self-Healing Theories

Self-healing theories are areas of research that seek to formulate arguments that explain the fundamental principles to be considered when implementing self-healing functionality and the pattern between self-healing and other areas of science. The self-healing cyber–physical system section describes what it means to have the self-healing functionality implemented into the cyber–physical system, and self-healing methods detail the models, frameworks, and network architectures that underpin the implementation of self-healing functionality.

Hence, different self-healing theories are presented and discussed in the following subsections.

### 2.1. Negative and Positive Selection

Negative and positive selection are two processes in the immune system to ensure that only healthy cells are present in the body. A self-healing system refers to a system that can repair itself when damaged or infected. Hence in the context of a self-healing system, the immune system uses both negative and positive selection to ensure that only healthy cells are present. If a cell is found to be harmful, the immune system eliminates it and then begins to repair and regenerate healthy cells. From the biological science viewpoint, negative selection is the process in which the immune system removes cells that recognise self-antigens, and the immune system uses negative selection to ensure that immune cells do not attack healthy cells. Likewise, positive is a process in which the immune system selects cells that recognise foreign antigens and prime the immune system to identify and eliminate harmful cells or pathogens. The CPS self-healing theory of negative and positive selection is the replication of the biological immune response in computer science. An example of such is using a genetic algorithm to detect system intrusions and then deploying the self-healing functionality of the algorithm to remediate the threat.

The characterisation of anomaly is essential in ascertaining where the potential threats or faults are located within a system, and the theory that is relied upon to achieve this is the negative and positive selection theory. Identifying threats before deploying practical self-healing functionality is a vital aspect of its implementation for appropriate remediation. Negative selection of anomaly detection is called "non-self" detection and positive selection

of anomaly detection is called "self" detection [13]. The central concept of negative selection, as shown in (Figure 2) is to construct a set of "non-self" entities that do not pass a similarity test with any pre-existing "self" entities. If a new entity is detected that matches the "non-self" entities, it is rejected as foreign.

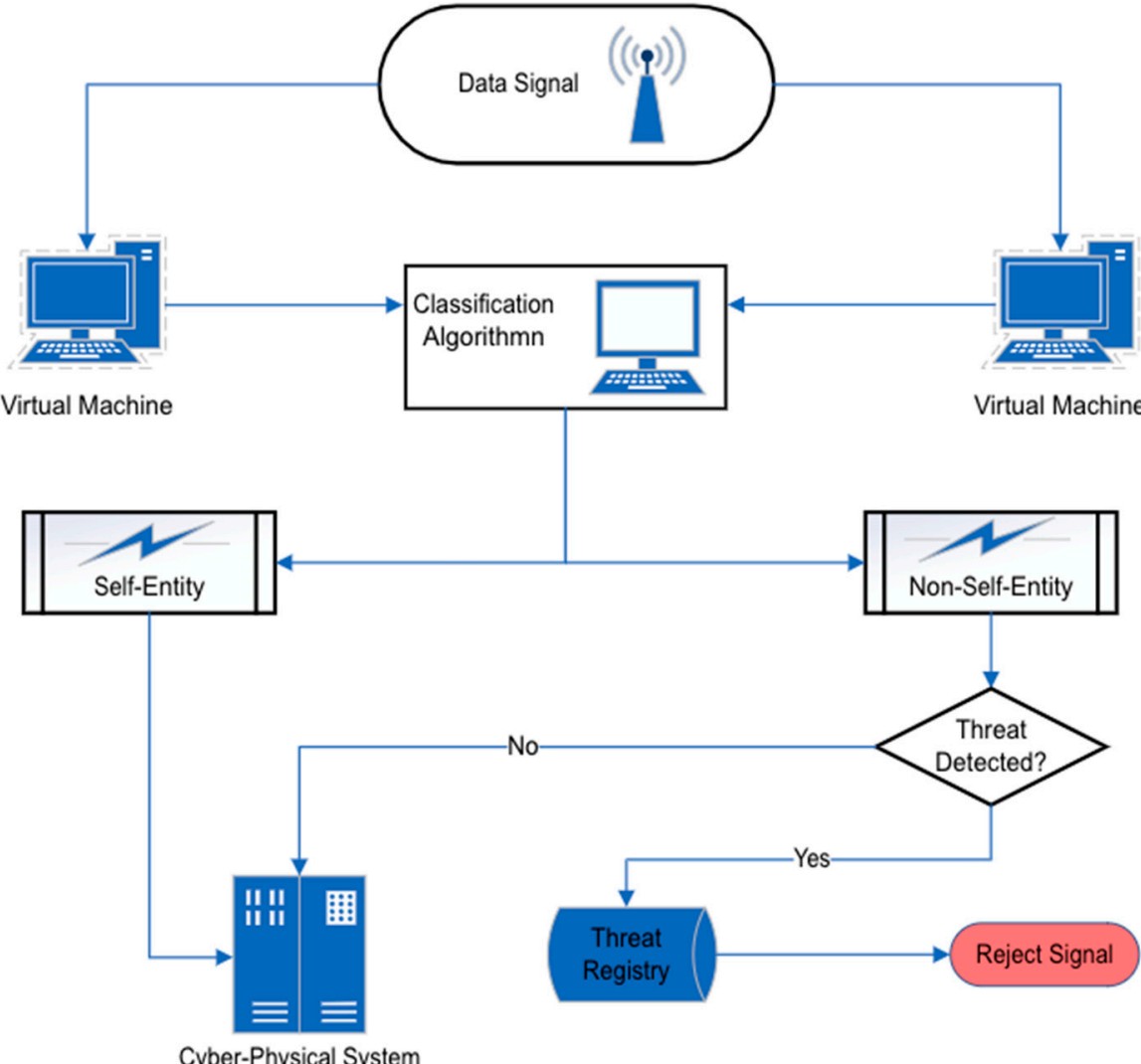

**Figure 2.** Negative and positive selection in self-healing systems.

Similarly, the positive selection principle reduces the algorithm by one step, and instead of matching a new entity with a constructed "non-self" entity set, it matches the entity with pre-existing "self" set and rejects the entity if no matches are found. D'haeseleer in [13] posits that negative selection has the properties of a thriving immune system, requiring no prior knowledge of intrusions. This is due to being, at its core, a general anomaly detection method. Negative selection is self-learning because it naturally evolves as a set of detectors; when obsolete detectors die, new detectors are obtained from the current event traffic. Dasgupta, cited in [13], argued that negative and positive selection produce comparable results despite their fundamental approach differences. Both approaches raise the alarm when an unknown entity infiltrates the system.

### 2.2. Danger Theory

Danger theory is the approach where immune responses are triggered by danger signals rather than just by the presence of any "self" or "non-self" objects. Negative or positive selection entities are allowed until signs indicate that they pose a threat. For

example, as shown in (Figure 3) within the immune system (which this theory is modelled against), if a harmful activity is detected, the immune response is triggered, attacking either all the foreign entities or entities locally, depending on the severity of the danger signal as noted in [13] that Burges et al. (1998) was among the first study that proposed the use of biologically inspired danger theory to detect and react to harmful activity in computer systems. Danger theory establishes the link between artificial immune systems and intrusion detection systems. Mazinger in [5] argued that danger theory is based on the concept that the immune system does not entirely differentiate between self and non-self but differentiates between events that possess the potential to cause damage and or the events that will not. Once the system understands itself, it can extend its pattern recognition capabilities and respond to dangerous circumstances.

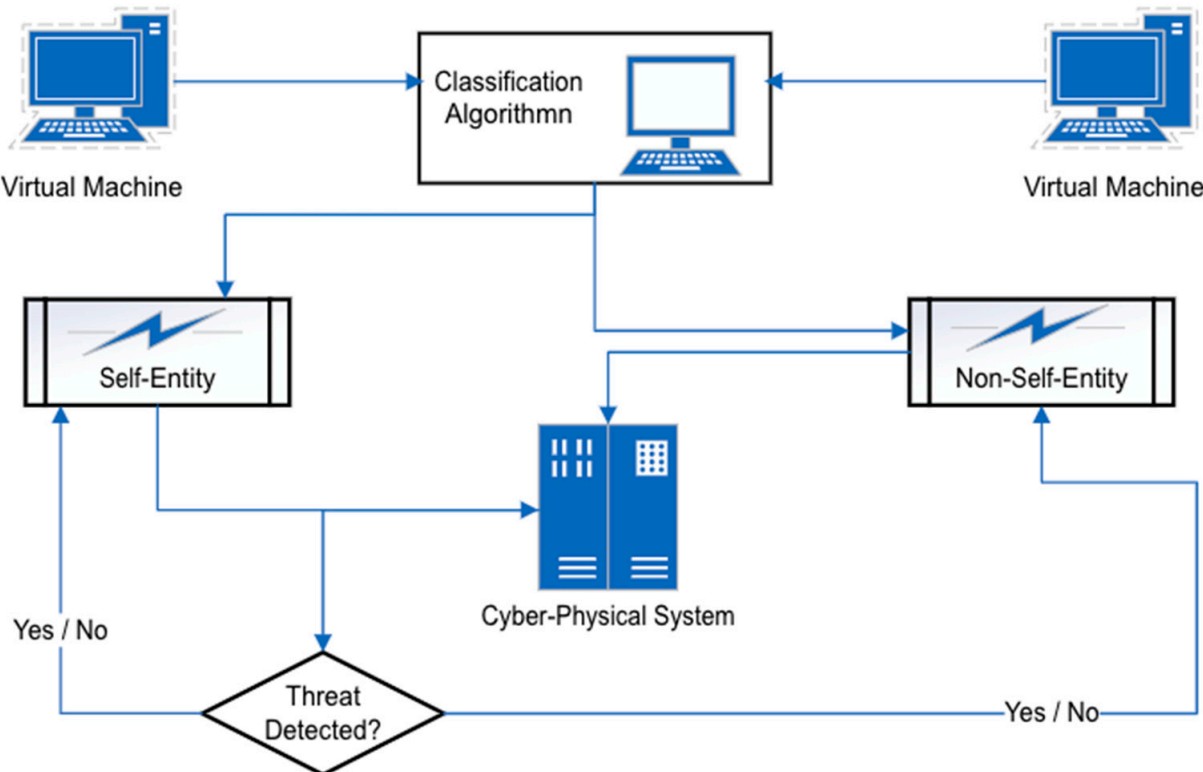

**Figure 3.** Harnessing the power of danger theory to optimise self-healing systems.

The creation of an intrusion detection self-healing system based on danger theory in which anomaly score is calculated for every event in the system was proposed by [13]. Each event has three computed values: event type (ET), anomaly value (AV), and danger value (DV). The ET is based on predefined types or automated events clustering. The AV defines how the abnormal event is based on "non-self" computations. The DV increases when any strange or potentially dangerous signal is associated with an event. All these three central event values are combined to calculate the threat total value (TV). TV is the perceived potential of a particular event to cause damage or to be a constitutional part of events that can cause a system's failure. Three main system flow originates from dangerous events [13]:

1. New event analysis: When a new event is detected, it should be added to the timeline, and the dangerous pattern should be checked;
2. Danger signal procession: When a danger signal is detected, the system must decide if any pattern can be related to the danger signal and then act accordingly;
3. Warning signal processing: When a warning arrives from other hosts that carry information about a danger signal and related dangerous sequence of events, a host's

timeline should be checked to verify that it does not have a similar dangerous sequence of events.

*2.3. Holistic Self-Healing Theory*

The holistic self-healing theory is a holism principle that reinforces complex systems' resilience. Improving the resilience of one part of the system can potentially introduce fragility in another. This occurs because when one aspect of the system's resilience is enhanced, it may inadvertently compromise the stability of another element. In mobile network management, for instance, Ref. [10] argued that this approach, as depicted in (Figure 4) means that different management domains and levels are not considered in isolation. Though the other management domains may be operating on different time scales and different managed objects, the domains need to be aware of the threat events that occur in each segment of the whole to react to the danger and trigger appropriate remedial action. Effective communication between the various subdomains of the system allows for the application of danger theory to protect the overall design as a singularity.

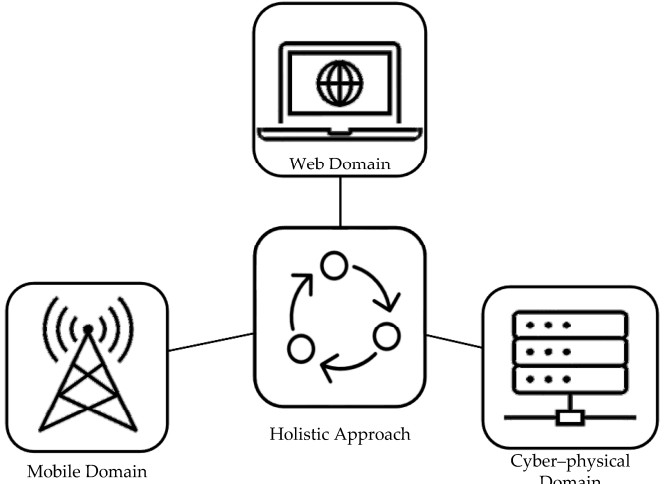

**Figure 4.** A holistic approach to maintenance and repair of the self-healing system.

## 3. Self-Healing for Cyber-Physical Systems

Alhomoud described a self-healing system in [13] as a resilient system that can carry on its normal functions even when under attack. A self-healing system is equipped with measures to identify and prevent attacks from internal or external events and to facilitate the system's recovery autonomously. A system equipped with self-healing functionality monitors the system's environment by constructing a pattern of the sequence of the events and using the pattern to detect anomalies in the circumstances before the remedial functions that correct or eliminate the events anomaly can be successfully deployed. Only when this autonomous remediation of attacks has been successfully achieved can the system be described as having demonstrated self-healing functionality. The main characteristic of a self-healing or self-organising system is the ability to react to problems through self-adaptive principles, which is shown in [19] using a platform they termed PREMiuM. The system must be able to classify the attack from everyday activities and take remedial actions to mitigate the impact. The proposed PREMiuM platform in [19] is designed to realise self-healing functionality in manufacturing systems, focusing on increasing efficiency during manufacturing processes. The PREMiuM platform consists of a top-level architecture of several services, i.e., interactive, self-healing, proactive, communication, modelling, and security services. These services, which are independent of each other, are deployed to achieve predictive maintenance of manufacturing systems. The self-healing service can detect or predict failures in the system in furtherance of the self-healing and self-adaptive functionality. A proposed intrusion detection system (IDS) by [13] is based on anomaly

attack detection in which the IDS monitors the system's environment, constructs a pattern of events and then uses the pattern to detect anomalies in the system, sometimes called outliers. The intrusion detection system detecting outliers triggers the system's defence mechanisms. Then, it notifies the other parts of the system and or system administrators of the anomalies that have been detected. In a similar self-healing approach, Ref. [1] proposed using machine-learning (ML) algorithms to implement IDS in smart grid construction by integrating traditional power grid strategy with the computer network. It then established the distribution fault-solving strategy library, which caused the grid to become self-adaptive. The method abstracted the power grid into an integrated domain with the cyber–physical system through data sharing, and the grid state in each of the system's nodes corresponds to a twin matrix, making the grid fully modelled and digitised. The grid, therefore, in the event of failure, utilises the fault-solving strategy library to self-correct itself using the functions of the distribution network. The critical characteristics of self-healing are reliability, fault tolerance, and flexibility. These characteristics are demonstrated in [27] principles of self-adaptation systems research, in which self-healing forms part of the fundamental principle encompassing self-protection, self-configuration, and self-optimisation.

RADAR, a self-healing resource, was evaluated by [12] using a toolkit called CloudSim, and the experiment results show a promising outcome, with an improvement in the fault detection rate of 16.88% more than the state-of-the-art management techniques. Resource utilisations increase of 8.79% is shown, and throughput increased by 14.50%; availability increased by 5.96%; reliability increased by 11.23%; resource contention decreased by 6.64%; SLA breaches of QoS decreased by 14.50%; energy consumption decreased by 9.73%; waiting time decreased by 19.75%; turnaround time decreased by 17.45%; and lastly, execution time reduced by 5.83%. The critical contributions of RADAR are listed in [12]:

1. Provision of self-configuration resources by reinstalling newer versions of obsolete dependencies of the system's software and offers management of errors through self-healing;
2. Automatically schedules resource provisioning and optimises QoS without the need for human intervention;
3. Provides algorithms for four-phased approaches of monitoring, analysis, planning, and execution of the QoS values. These four phases are triggered through corresponding alerts to aid the preservation of the system's efficiency;
4. Reduces the breach of service level agreement (SLA) and increases the QoS expectation of the user by improving the availability and reliability of services.

A prominent issue in current research is the ability of systems to identify "zero-day" or never-before-seen anomaly events intelligently. The proposal presented by [33] suggests utilising a knowledge-based algorithm to construct an intrusion detection system (IDS) that effectively prevents power grid fault line intrusion. Experiments were conducted within a testbed of a six-bus mesh network modelled to identify fault events within the system and concurrently perform mitigating actions initiated by [33] and proposed as a novel protocol. The proposed protocol is referred to as autonomous isolation strategies. The strategies involve rerouting power flow displacements within the power grid once a threat intrusion is detected. Simulations during the experiment were conducted using Power World Simulator, MATLAB, and SimAuto (a fault detection platform). As noted in [33], the experiment result shows that MATLAB extracts network parameters. Then, self-healing strategies are triggered by rerouting network processes to other distribution areas, providing stability to the system. The self-healing approach, started by the knowledge-based algorithm, continues concurrently until all the overloading lines on the grid are cleared and all effects of the system's threat eradicated. The guidelines of supervised learning for the knowledge-based algorithm as related to the electric power network are listed as having the following characteristics in [33]:

1. Detection of overloaded transmission lines in the power network;
2. Identify buses that have overloaded transmission lines connected to them;

3. Identification of the busbar that has the highest reserve capacity and that can then serve as a viable option for a power restoration strategy;
4. Identification of the nearest distribution generator to the overloaded transmission line;
5. Identification of the termination point of the overloaded lines;
6. Establishment of line connection using the references of the reserve busbar index.

Other anomaly detection systems for network diagnosis are proposed in previous studies, such as ARCD by [34], which uses data logs collected from large-scale monitoring systems to identify root causes of problems in a cellular network. An experiment by [34] identified that ARCD systems achieved rate levels above 90% in terms of anomaly detection accuracy rate and detection rate. The drive towards an automatic diagnosis of computer systems failures in mobile cellular networks is propelled by the industry's need for efficient means of identifying problems within the network. Interestingly, Ref. [35] noted that mobile network operators spend a quarter of their revenues on network maintenance, and a drive towards maintenance automation will drive down costs. A solution that relies on random tree forest (RF), convolutional neural network (CNN), and neuromorphic deep-learning module to perform fault diagnostics were proposed by [35]. The proposal uses an RSRP map of fault-generated images to provide an AI-based fault diagnostic solution. The impact of fault diagnostic solutions is noticeable in reducing costs and improving the end user's overall quality of service (QoS). Experiments during research by [35] show that the proposed system could identify all the faults fed through the image datasets.

Similarly, a system that is resilient to system intrusions and built using Python-based libraries, software-defined networks, and virtual machine composition was proposed by [3]. The system is called Shar-Net and was tested in a smart grid environment. The experiment results show demonstrably viable IDS that can prevent cyber-attacks and, at the same time, can mitigate the effects of attacks through the system network's automatic reconfiguration. The principal areas covered in the proposed system are intrusion detection system (IDS), intrusion mitigation system (IMS), and alert management system (AMS). Zolli and Healy describe the resilience of a system in [10] as the ability of the system to recover from failure or attack. The above description is quite different from a robust system. A robust system is a system that is built to withstand unforeseen threats. Although the two terms describing the core functionality of a self-healing-capable system might be used interchangeably, it is essential to note the difference between them. A robust system relies on threats that have been previously seen and thus has allowed the designers of the system to build countermeasures to such threats proactively. On the other hand, a resilient system retroactively reacts to unforeseen or "zero-day attacks" and applies countermeasures accordingly. The authors of [10] listed the following principles of a resilient system:

1. Monitoring and adaptation: It must be responsive to unforeseen attacks;
2. Redundancy, decoupling, and modularity: It must have a decentralised structure to prevent the threats from spreading to the other constituent parts of the network or the system's host;
3. Focusing: The system must be able to focus resources where they are most needed to prevent the overuse of resources, which may be counterintuitive to the task of shoring up the system's resilience;
4. Diverse at the edge and simple at the core: The system should be able to utilise shared protocols through simply defined processes. Still, it should also retain an element of diversity to circumvent widespread attack threats.

Self-healing functions can be implemented in four stages (Figure 5). These include profiling the system's normal states, detecting the system's deviation from its normal state, diagnosing the system's failures, and taking corrective actions to mitigate the impact of the system's failure. The choice of profiling algorithm for a self-healing system is dependent on the scope of the design requirements and based on further considerations such as:

1. The system's architecture;
2. The available datasets;

3. Profile scope;
4. Profile features;
5. Feature distribution or subset;
6. Understandability.

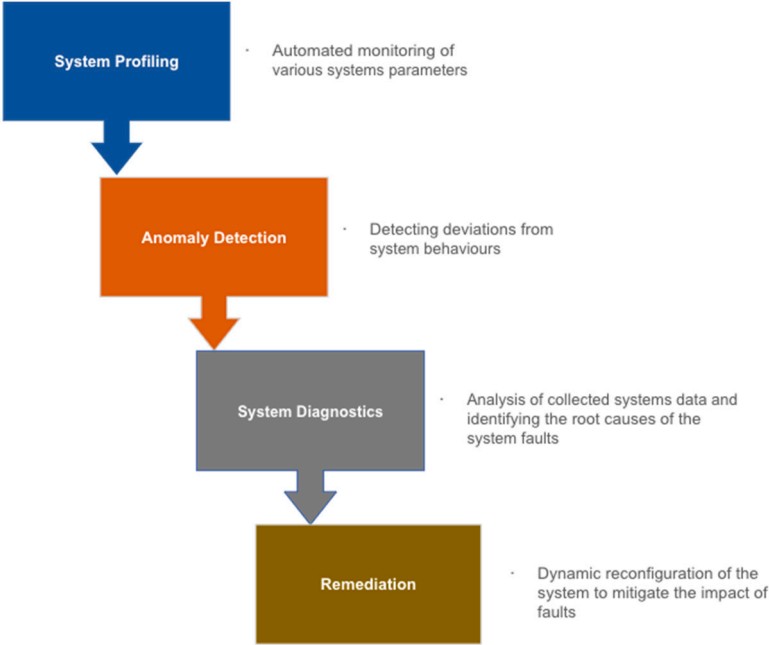

**Figure 5.** Four stages of a self-healing system: implementation and functions.

The illustration of anomaly detection and diagnosis for radio access networks (RANs) (Figure 6) shows the profiling, detection, and diagnosis of anomaly events related to RANs. The self-healing function is implemented according to a selected event context, like when a threat event occurs. The key performance indicators (KPIs) are calculated when a profile is created within a time-series format. The anomaly events that are unique in characteristics are detected based on their anomaly level. The diagnosis function then analyses the detected anomaly occurrences. The diagnosis function then identifies the root causes of the anomaly events to ascertain whether corrective measures are required or not to lessen the potential threats, and the corrective workflow is then triggered if indeed needed [10]. The major problem that affects the optimal performance of the smart grid network today is the occurrence of system failures caused by multifaceted fault areas, such as system overload, system intrusion, and system misconfiguration, among others [1]. Such failures within the smart grid can cause significant economic setbacks, with consequences that sometimes negatively impact human livelihoods or quality of life. To mitigate the problem of the system's inability to self-heal after failures, Ref. [1] proposed using a fault-solving strategy library based on a twin model system and machine learning (ML) algorithm to implement a self-healing mechanism in a smart grid. The algorithm will be fed into the dataset derived from the fault-solving library to detect anomalies within the system. Then, the self-healing function is trigged once the classification process is completed and a viable mitigation solution is found. Self-healing methods can be helpful in a variety of contexts where uptime, reliability, and performance are critical, such as:

- Manufacturing: In a manufacturing environment, production lines and equipment must always be operational and available to ensure maximum output. Self-healing mechanisms can detect and respond to faults or failures automatically, thereby minimising downtime and reducing the need for manual intervention;
- Transportation: Transportation systems, such as trains, planes, and automobiles, rely on sensors and other technology to monitor and control their operations. Self-healing

mechanisms can detect faults or failures and take corrective action to ensure the system's safety;

- Power grids: Power grids are critical infrastructure that must always be operational to ensure reliable access to electricity. Self-healing mechanisms can detect and respond to faults or failures, preventing cascading failures and reducing the impact of outages;
- Healthcare: Healthcare systems rely on technology to monitor and provide critical care. Self-healing mechanisms can ensure that these systems are always operational, minimising the risk of disruption that could compromise patient safety;
- Internet of Things (IoT): IoT devices are becoming increasingly common in homes, businesses, and public spaces. Self-healing mechanisms can detect and respond to faults or failures, ensuring these devices remain operational and connected to the Internet.

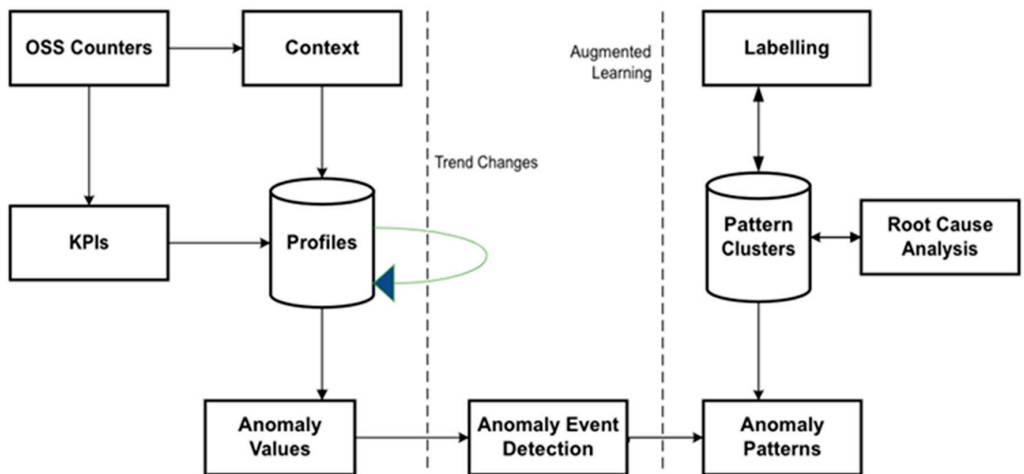

**Figure 6.** Anomaly detection for radio access.

## 4. Self-Healing Approaches

Self-healing approaches are the strategies and techniques to detect, diagnose, and resolve CPS problems automatically without human intervention. These approaches are commonly used in complex systems such as software applications, networks, and hardware systems to ensure that they recover from failures and continue to operate without disruption. Table 2 lists the most common machine-learning tools for cyber–physical self-healing systems.

The approaches involve:

- Redundancy: Redundancy involves having duplicate components or systems that can take over if the primary system fails. For example, if one node fails in a computer cluster, another node can take over and continue processing the request;
- Automated recovery: Automated recovery involves setting up automated processes to detect and resolve problems. For example, a computerised process can restart a server or move its workload to another server if it goes down;
- Predictive maintenance: Predictive maintenance involves using sensors and data analytics to predict when a system is likely to fail and proactively take action to prevent the failure from occurring. For example, an aircraft engine can be monitored for signs of wear and tear, and maintenance can be scheduled before failure occurs;
- Machine learning: Machine learning involves using algorithms to analyse data and learn patterns that can be used to detect and resolve problems. For example, machine algorithms can analyse network traffic and detect anomalies that may indicate security breaches;
- Fault tolerance: Fault tolerance involves designing systems that can continue to operate even if one or more components fail. For example, a database cluster can be designed

to replicate data across multiple nodes. If one node fails, then other nodes can continue providing data access.

**Table 2.** List of machine-learning tools for cyber–physical self-healing systems.

| | | |
|---|---|---|
| Self-Healing Machine-Learning Tools | Model and Framework | Twin Model |
| | | QoS Model |
| | | Auto-Regressive Moving Average with Exogenous Input Model |
| | Network Architecture | Strategy Network |
| | | Valuation Network |
| | | Fast Decision Network |
| | | Intrusion Detection System |
| | | Phasor Measurement Unit |
| | | Agent Architecture |
| | | Host Intrusion Detection System |
| | | Multi-Area Microgrid |
| | Algorithms | Monte Carlo Tree Search |
| | | Artificial Neural Network |
| | | Supervised Knowledge Base Algorithm |
| | | Genetic Algorithm |
| | | Dynamic Detection Algorithm |
| | | Support Vector Machine |
| | | Naive Bayes |
| | | Random Forest |
| | | DBSCAN Algorithm |
| | | Long Short-Term Memory (LSTM) |
| | | Auto-Regressive Moving Average (ARMA) |

It is important to consider training datasets, which are the bedrock of implementing ML self-healing functions in cyber–physical systems. The self-healing algorithms use data to identify errors, analyse their causes, and take remedial actions. Self-healing algorithms utilise the system's derived data to improve the reliability of the system and fulfil the self-healing functionality, data such as the following:

- Log data: This contains information about the system events, such as error messages and other data that can be used to diagnose problems;
- Performance metrics: This includes data derived from CPU utilisation, memory usage, network latency, and input/output disk;
- Configuration data: Includes data related to the system's configuration changes and parameters;
- Environment data: Identifies the issues relating to environmental conditions, such as overheating and excessive humidity.
- User behaviour data: These data identify patterns in the system's user behaviours, such as the response times or the frequency of errors.

### 4.1. Self-Healing Models and Frameworks

A self-organising network was presented by [10], called the self-organised network (SON) experiment framework, with which self-healing functionality is implemented and demonstrated using data from actual network instances and live integrations. The SON experiment framework is a tool developed by Bodrog [10]. It is implemented in R language and has a user interface to visualise anomaly detection and its diagnosis process. Further research includes using the transfer-learning method to investigate the remediation aspect of the self-healing system. A reoccurring theme is addressed in the literature relating to self-healing methods, and [36], in describing self-healing software techniques, noted that the techniques are modelled after the observer orient decide act (OODA) feedback loop. The OODA model identifies where to apply protection by observing the system's behaviour. The system is monitored to detect the fault and determine fault parameters, such as the type of fault, the input or the sequence of events that led to the input, the approximate areas of the system that is affected by the defect, and the information that may be useful in mitigating the fault. The self-healing mechanism in terms of OODA is more appealing than the traditional defence mechanism, which prioritises the termination of attack processes and restarting the system in the event of an attack. The self-healing tool succeeds by preventing code injection or the misused of legitimate code, rather than the traditional defence method, which may cause systems fault to persist even after the attack process has been terminated and the system restarted.

Self-healing functionality implementation has at its core anomaly detection, and once an anomaly is detected using the various methods that are presently available, such as network intrusion detection system (NIDS) and or host-based intrusion detection systems (HIDS), then performing remediation through triggered actions becomes necessary to protect the system and realise the self-healing function. A self-healing system must encompass resilience through its ability to take corrective measures that return the system to its routine or default state after system failure or attack. Systems that are capable of remedial actions against losses and or attacks from outside sources were proposed by Lui in [37]. However, the proposal cannot trigger automatic remediation to mitigate attack events in real time but still requires human intervention to perform the remediation process. Therefore, a self-healing framework that is automatic and uses collaborative host-based learning to incorporate a self-healing mechanism into the Internet of Things (IoT) devices was proposed by Golomb in [37] in a study where a lightweight HIDS designed for IoT was deployed. The authors of [33] described the conventional self-restoration within the electrical systems concept as a system that can automatically reconfigure itself to achieve repossession during an unexpected power disruption event. Automation within the power grid is the concept of autonomously restoring proceedings that are impacted by power failures and restoration of power supply through redirecting power flow from stable lines to the fault-affected areas instantaneously.

The automation principle facilitates reforming the existing power grid to dynamically respond to fault detection, applying deterministic isolation techniques and executing reroute operations in real-time. An example is shown in a framework demonstrated by [37] in the form of a tool for detecting anomaly events on cyber–physical systems. The system sends alerts through the HIDS and triggers the best possible remediation action, neutralising the attack effects and returning the IoT devices to their normal state. An auto-remediation model is deployed, which uses an evolutionary-computation algorithm built to imitate the functions of the natural process of evolution and in which the fittest are likely to survive through a process like natural selection. The principle of the evolutionary algorithm dictates that multiple practical solutions are created, and the best among the solutions is then selected during the evolutionary process. Consequently, various solutions for the different ML models implemented on IoT devices connected to local area networks (LAN) are defined.

The best model, which becomes more adept at selecting the correct remediation process, is chosen more often than the rest. The evolutionary algorithm uses fitness criteria,

which enables the selection of models based on the health score provided by the algorithm. In addition to selecting the models based on the health score, the algorithm generates new ML models based on the individual model's current attributes and supported by mutation. To initialise the automatic remediation models, Ref. [37] proposed the implementation of a lab setup in which multiple versions of auto-remediation models are trained to provide corrective countermeasures by classifying system attacks that can be deployed on IoT devices. The approach expedites the collaborative training process and improves the ability of the auto-remediation agents to trigger corrective countermeasures against actual attacks within a real system environment.

A physical platform comprising 35 Raspberry Pi devices, with similar hardware and software on each device, was used in experimentation. All the Raspberry Pi devices were connected to a network switch on a LAN. The long short-term memory (LSTM) model was initialised on each IoT device with random weight to evaluate the experiment. Then, an attack is triggered on the devices using attack stimulators based on the Red Hat Ansible engine, an IT automation tool. After each episode, the genetic algorithm is executed to update the LSTM model through learning iteration. The learning iterations are measured on all the devices connected to a testbed against the response of the attack, and a countermeasure action is undertaken to return the devices to their normal states. One of the critical discoveries during the experiment is that increasing the number of devices connected to the testbed (i.e., training sets) decreases the learning process time exponentially, as shown in (Figure 7).

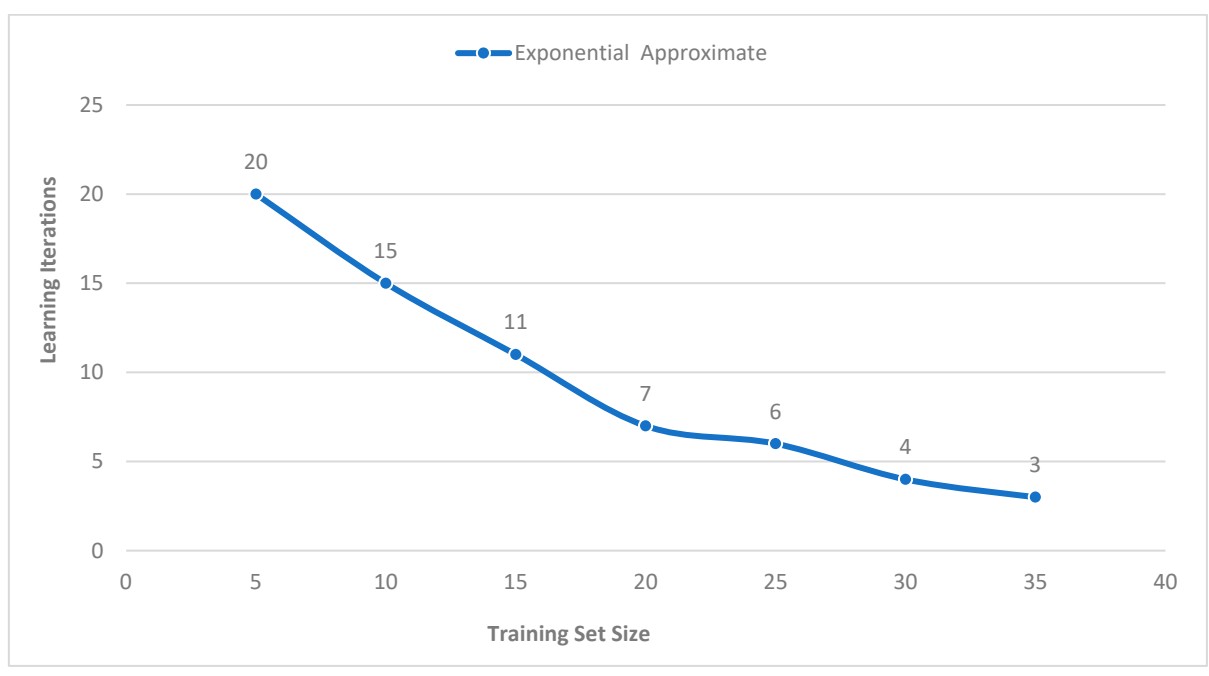

**Figure 7.** Learning iteration of long short-term memory model (LSTM).

In presenting their work, Goasduff [37] noted that Gartner Inc. predicted that there would be 5.8 billion IoT devices by the end of 2020, and the prediction represents a 21% increase from the previous year. The number is then expected to reach 64 billion devices by 2025. Theoretical studies show that ML models, learning autonomously by experience and collaboration, can serve as the basis for new cyber systems security defence. Future studies could see the automatic remediation algorithms built into IoT devices that could provide self-healing system safeguards and reduce maintenance costs, which can ordinarily occur from system failure or attack. Future work to further explore the theory would be centred around adding more classification of attacks to the ML models to bolster the

learning capacity of the models, as well as explore the feasibility of transferring a trained automatic remediation model between multiple IoT devices.

Similarly, a self-healing function within the microgrid electricity network was described by Shahnia in [26] as part of a control and operation mechanism aimed at introducing a level of automation in the network system. The self-healing mechanism involves many decision-making processes and can be presented in a three-tier hierarchical structure. The bottom part of the structure consists in deploying a well-established control scheme strategy to solve fault problems locally without requiring interventions from any central command. The goal of the strategy is to reduce costs and increase the speed of operation. According to Wang and Wang [26], the goal prompted many researchers to explore local decision-making models.

Consequently, Wang and Wang [26] proposed a novel sectionalised self-healing approach in decentralised systems to resolve electrical energy distribution problems. The objective set by the authors is to guarantee power supply by balancing loads in the subsection of the decentralised system by either adjusting the power outputs of the distribution network or through the implementation of load shedding. A suitable self-healing network deployment case study is a platform by [20] called fault location isolation and supply restoration (FLISR), deployed and tested on a network simulator. The test result indicates that the solution reduces the cost of commissioning on grid networks, and the platform has been deployed in grid networks in several countries, such as the Netherlands, France, Vietnam, and Cuba. A method for an automatic prediction model for systems failure, recovery, and self-healing in virtual machine (VM) networks using intelligent hypervisors was presented by [9]. The self-healing functionality implementation on the transmission network of a smart grid is achieved through optimal voltage control with a genetic algorithm, unified power flow controller (UPFC), and islanding process. The distribution network approach involves the design based on the propulsion system, ant colony algorithm, multi-stakeholder control system (MACS) for the intelligent distribution network, fault location, isolation, and service restoration (FLISR). The self-healing functionality is achieved using a predictive model that recovers failed VM instances in a physical machine through a self-healing algorithm that utilises the VM's memory snapshots. The self-healing algorithms include decision tree, Gaussian normal basis (GNB), and support vector machine (SVM). A database's self-healing functionality, extracting helpful information that identifies the type and subtype of events from the database's events report (text fields) data, was presented by [38]. The extracted information was analysed using Python version 3.6.3, Natural Language Toolkit (NLTK), Scikit-learn, Regular Expression (RE), and Pandas modules. Several self-healing methods have been proposed, such as those described in [16] and relating to a smart grid with corresponding characteristics, which has the elements and is representative of all self-healing known features. The smart grid self-healing systems, to be considered adequate, must be able to meet the following criteria:

1. Quick detection of system faults;
2. Redistribution of network resources to protect the system;
3. Reconfiguration of the system to maintain service, irrespective of the situation;
4. Minimal interruption of service during reconfiguration or self-healing period.

### 4.1.1. Twin Model

The twin model is a statistical model commonly used in the behavioural genetics study of the heritability of various traits and behaviours. It assumes that genetic and environmental factors contribute to the variation in each trait or behaviour. Researchers may use the twin model as a statistical model and ML algorithms to analyse data from sensors and other sources to detect and diagnose anomalies in a cyber–physical system. The concept of the twin model to this effect was proposed by [1], and it integrates a strategy network, valuation network, and fast decision network into the twin matrix of the power grid. It facilitates the analysis of the smart grid by operating the actual power grid but does so virtually (on a computer). The model is a utility used to simulate the operation of an

existing grid on a computer, with a use case that analyses the functionalities of a smart grid. As merging energy flow and information become vital for the optimal operation of the smart grid, failure in these aspects can cause successive losses in the transmission between data and the physical network of the power grid. There may also be a resultant effect from such failures that precipitated the collapse of the computing system and the entire smart grid. To better describe the grid state, Ref. [1] introduced a virtual network called the twin model, which represents the physical system state and the corresponding data in precise information in the smart grid. The relationship of which is defined in the following equation.

$$A = \begin{array}{l} A_1\ B_1\ C_1 \longrightarrow N_1 \\ A_2\ B_2\ C_2 \longrightarrow N_2 \\ --------------------- \\ A_n\ B_n\ C_n \longrightarrow N_n \end{array}$$

The data of each node in the grid are compared to the matrix, and each node in the grid is regarded as the first column of the matrix, which is noted as $A_2 A_3$ the different types of faults (voltage, frequency, etc.) may appear in the subsequent corresponding node, which is noted as $B_1\ C_1$ etcetera. Through the above representation in the actual grid, if a node has a fault, the failed node can easily be identified by analysing the data changes in the twin matrix.

### 4.1.2. QoS Model

The quality of service (QoS) model is described by [17] as the expression of the correlation between QoS changes and its environment primitives (EPs) or control primitives (CPs). The QoS models can be a powerful tool to automatically assist cloud providers in adapting cloud-based services and applications. They help determine the extent to which services and applications can sufficiently exploit CPs to support QoS objectives and consider the QoS sensitivity of both EPs and CPs, as noted by [37]. A self-healing QoS model can automatically detect and correct errors in a system without human intervention. It is a model designed to maintain high accuracy and reliability despite unexpected events or changes. One approach to building a self-healing ML QoS model combines supervised and unsupervised learning techniques. Supervised learning is used to train the model on a set of labelled data, while unsupervised learning is used to identify patterns and anomalies in the data.

### 4.1.3. Auto-Regressive Moving Average with Exogenous Input Model

The auto-regressive moving average with exogenous input (ARMAX) model is a statistical model that combines both auto-regressive (AR) and moving average (MA) components with exogenous input. Ref. [2] explored combining artificial neural network (ANN) and auto-regressive moving average with an exogenous input model (ARMAX) to show how primitives correspond to the quality of service adaptively (QoS) based on the related primitive matrix. Ref. [2] through experiment demonstrate the implementation of a middleware that incorporates a self-adaptive approach based on the feedback control mechanism. The experiment's outcome, when tested using the RuBis benchmark and FIFA 1998 dataset, proves that the models (ANN and ARMAX) produce a more accurate result when compared to the state-of-the-art models. The resulting model from combining the two models in an experiment is S-ANN and S-ARMAX. The former handles the dynamic QOS sensitivity better and produces higher accuracy in events where QOS fluctuations occur, whereas the latter makes fewer errors when QOS fluctuations decrease. The ARMAX model is commonly used in time-series analysis to predict future values tina time series based on past values and exogenous inputs. The model can be estimated using various methods, such as maximum likelihood estimation or least square estimation, and evaluated using measures such as the Akaike information criterion (AIC) or Bayesian information criterion (BIC).

### 4.2. Network Architecture

The principal challenge of research into self-healing functionality on computer networks is the ability to achieve reliability, one of the three core characteristics of a self-healing system described [27,29]. A method was proposed based on utilising shared operation and spare nodes in each neural network layer to compensate for any faulty node and resolve the self-healing reliability challenge. The proposed method is implemented using VHSIC hardware description language (VHDL), and the simulation result is obtained through Altira 10 GX FPGA. The experiment by [21] looked at overcoming the area overhead caused using redundancy over time, and the result demonstrates overhead reduction by 27% for four nodes within a layer and a 15% reduction in overhead for ten nodes within a layer. When designing self-healing systems, several network architecture issues need to be taken into consideration:

- Redundancy: The network should be designed with redundant components to minimise the impact of failures by providing fail-safe functionality to the system;
- Automation: The network should be automated to reduce the need for manual intervention and speed up the recovery process;
- Monitoring: The network should have robust monitoring capabilities to detect and diagnose issues as soon as they occur;
- Resiliency: The network should be designed to be resilient to common failures, such as power outages, hardware failures, and software bugs;
- Security: The network should be designed with safety in mind to mitigate attacks that could cause outages

The above will ensure that the three main self-healing characteristics (Figure 8), as defined by [16], can be realised.

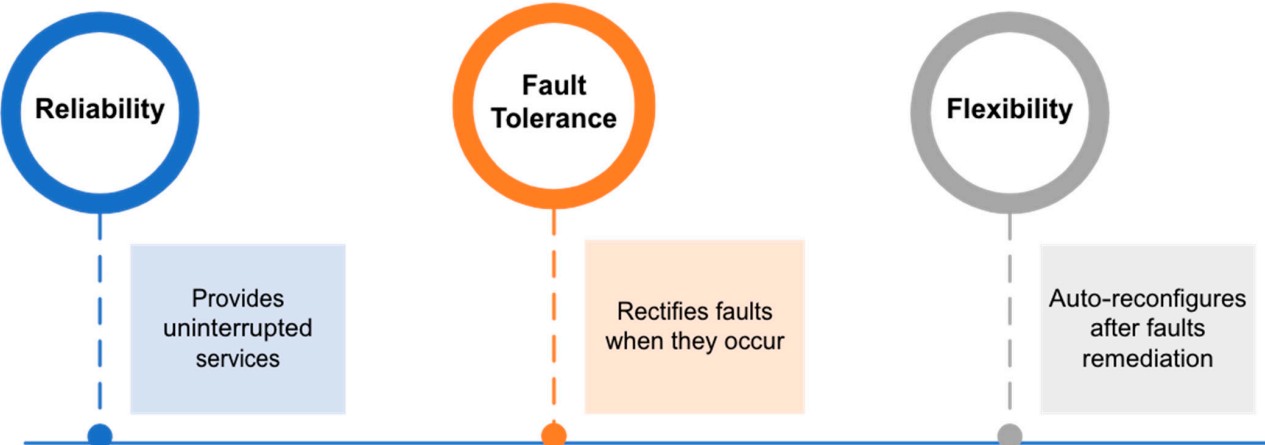

**Figure 8.** Characteristics of a self-healing system.

Having evaluated self-healing approaches within the cluster platform architecture, Ref. [14] proposed a self-adaptive method that identifies and recovers from abnormal events in cluster platforms such as Kubernetes or Docker. The proposed method is designed to introduce different anomalies into the cluster architecture of an edge computing platform. An experiment was conducted using generated workloads data to measure the effects of the abnormalities on the edge computing nodes at varying system settings, which allowed for identifying relationships between the system components and aided the system's adaptation towards fulfilling its overall self-healing goals. The experiment result shows that the proposed approach detects anomalous events with accuracy ratings of 98% and recovery ratings of 99%. In evaluating self-healing in virtual networks (VN), Ref. [29] presented a multi-commodity flow problem (MFP) network called MFP-VNMH, which can enhance the VN mapping of overall VNs. The proposed approach enables self-healing capabilities on network virtualisation. Self-healing functionality was achieved using sessions between

the service pack (SP) and inhibit presentation (InP), supervised by a collection of control protocols. Experiments conducted on the proposed approach demonstrate efficiency and effectiveness for service restoration after network failures.

### 4.2.1. Strategy Network

A network comprising tens of thousands of faults solving strategy libraries was proposed by [1]. They argued that the essence of the strategic network in their study is to proffer a combination of solutions when dealing with risks of power grid failures. The global strategy network in a power grid scenario is not accurate in solving power failure problems; hence, the sub-strategy of each node is combined to solve a global fault based on the general direction of the global strategy. The combined sub-strategy then constitutes a strategy network. The framework for the network consists of several layers and each with its strategy:

- Perception layer: This layer monitors the network to detect failures and anomalies. Strategies in this layer include using sensors to monitor network traffic, analysing system logs, and applying ML techniques to identify abnormal patterns;
- Analysis layer: This layer is responsible for analysing the data collected by the perception layer to identify the root cause of failures. Strategies in this layer include using ML algorithms to analyse the data and identify patterns that indicate the cause of a failure;
- Planning layer: This layer is responsible for developing a plan to address the identified failures. Strategies in this layer include using ML algorithms to determine the optimal recovery strategy and selecting the appropriate recovery mechanism;
- Execution layer: This layer is responsible for executing the recovery plan. Strategies in this layer include using automation to execute the recovery plan and providing feedback to the other layers to optimise the recovery process;
- Knowledge layer: This layer stores and manages knowledge about the network and the recovery process. Strategies in this layer include using databases to store information about the network topology and previous failures and using algorithms to learn from previous failures and improve the self-healing process.

### 4.2.2. Valuation Network

In artificial intelligence and deep learning, a valuation network refers to a neural network model designed to estimate or assign a value to a particular input or set of information [1]. The purpose of a valuation network is to evaluate the quality, significance, or relevance of the input data based on a specific criterion or objective. Valuation networks are commonly used in various applications, such as reinforcement learning, where the network is trained to estimate the value or expected return of different actions or states in an environment. They can also be used in recommendation systems to assess the preference or utility of items for a user or in natural language processing tasks to assign a score or sentiment to text inputs. The architecture of a valuation network can vary depending on the specific task and requirements. It typically involves multiple layers of neurons that process and transform the input data, and the final output represents the estimated value or score assigned to the input. The valuation network solves the problem of configuring the local strategy by judging the sub-strategy effective solution rate under specific power grid scenarios. At its core are machine and data learning. The practical solution rate means the combination of sub-strategy configuration, considering the global impact. The valuation network, Ref. [1] argued, is not predisposed to guess what sub-strategy to take but relies on an optimal global angle to predict the effective-solution rate of being resolved under different local strategy configurations or effective probability. The valuation network is a vital component of the continuous improvement of the fault-solving strategy library. By analysing the network's usage patterns, the valuation network can identify areas where resources are underutilised or over utilised and recommend adjustments to improve and reduce costs. Valuation networks can help improve the

system's reliability, scalability, and cost-effectiveness by continuously monitoring the performance and optimising resource utilisation.

### 4.2.3. Fast Decision Network

In the context of self-healing, a fast decision network architecture can make quick, accurate decisions to enable rapid failure recovery. Speedy decision-making is critical to minimise downtime and maintain system availability in a self-healing system. A fast decision network can be implemented using ML algorithms to quickly analyse network data and make decisions based on real-time information. The algorithms can be trained to recognise network traffic patterns and behaviours and identify potential issues before they result in a system failure. For example, if a fast decision network detects that the CPU usage is high or that any other performance issues, it can quickly take action to mitigate the problem. This could involve diverting traffic to other devices, scaling up resources to handle the increased loads, or automatically triggering a self-healing mechanism to recover from the failure. A fast decision network is described by [1] as a representation of the amount of decision-making speed. It starts from the judging position and then quickly takes the local strategy to solve the grid risks for each node. The decision-making network will have a good or bad result after the configuration sub-strategy until the final node, and calculate the statistical probability of each corresponding node to obtain the overall probability. The process of a fast decision network does not consider the effect of local strategy configuration on global strategy. Fast decision network plays a significant role in enhancing the speed of fault resolution when an actual online operation is running. A fast decision network works more effectively when combined with a valuation network in a strategic configuration.

### 4.2.4. Virtual Machine

A virtual machine (VM) is a software emulation of a computer system that can run an operating system and application like a physical computer. In a self-healing system, VM can implement a self-healing mechanism in several ways. For example:

- Isolation: A VM can isolate applications and services from each other. If one application or service experiences a failure, it can be restarted within the VM without affecting the other applications or services running on the same physical machine;
- Redundancy: Multiple VMs can be deployed to provide redundancy for critical applications or services. If one VM fails, another can take over its workload to ensure continuity of service;
- Rapid provisioning: VMs can quickly be configured to meet changing workload demands. The proposed method enables the network to scale up or down as needed by utilising shared operation and spare nodes in each neural network layer, ensuring performance and availability are maintained;
- Testing and validation: VMs can test and validate self-healing mechanisms before they are deployed in a production environment. Implementing these approaches can help ensure the effectiveness of the tools without causing unintended consequences.

In their study, Ref. [17] investigated heterogeneous events in which different software stacks run on virtual machines (VMs) and physical machines (PMs) within a cloud environment, each operating at varying levels of primitives and capacity. The primitives tend to correspond to similar heterogeneous QoS for different service instances. Adaptive QoS models concerning each service instance were created to overcome the heterogeneity problem. As such, the service instance on VM (virtual machine) utilises the same computational resources as other service instances running on different VMs. Network virtualisation has been adjudged a novel approach for implementing promising heterogeneous services and building the next-generation network architecture [29]. Network virtualisation through VMs promises better utilisation of resources and increased network service delivery. Network virtualisation's possibilities prompted [29] to propose a novel VN restoration approach called MFP-VNMH, which enhances VN mapping and

service restoration. By isolating applications and services, providing redundancy, enabling rapid provisioning, and facilitating testing and validation, VMs can play a critical role in implementing effective self-healing mechanisms.

### 4.2.5. Phasor Measurement Unit

Phasor measurement unit (PMU) can be used in the context of self-healing to monitor and analyse the state of the power grid in real time. PMUs measure electrical quantities' magnitude and phase angle, such as voltage and current, at high speed and accuracy. PMUs can be used to implement self-healing mechanisms in several ways, such as:

- Real-time monitoring: PMUs can be used to monitor the state of the power grid in real-time, providing high-resolution data on voltage, current, and frequency. This data can be used to detect and diagnose faults and other anomalies in the grid;
- Fault detection and isolation: PMUs can detect faults in the power grid, such as short circuits or equipment failures, by analysing electrical quantities' magnitude and phase angle. PMUs can identify the location and the extent of the fault;
- Restoration: PMUs can be used to facilitate the repair of power after a fault has occurred. By providing real-time data on the state of the grid, PMUs can help operators quickly identify the source of the fault and take steps to restore power;
- Protection: PMUs can be used to protect critical equipment and infrastructure. By monitoring the state of the power grid in real-time, PMUs can detect abnormal conditions and trigger protective measures, such as tripping circuit breakers or isolating faulty equipment.

PMU for a self-healing feature on the power grid was implemented by [18] and created real-time monitoring and load balancing using three components that facilitate the self-adaptation and self-healing functionality of the network. The following list describes the three components of the PMU:

1. Intrusion detection system (IDS): Relies on the PMU network logs and phasor measurements to detect different classes of abnormal events within the network;
2. Intrusion mitigation system (IMS): Once the IDS detects an anomaly, the generated alerts from AMS are delivered through a publisher–subscriber interface; for appropriate remedial action to be taken;
3. Alert management system (AMS): Generates alerts based on anomaly rules defined in IDS and forwards the alert to the IMS if abnormal events are detected for onward remedial actions by the IMS.

AMS comprises three sub-components: the alert manager subscriber1, subscriber2, and subscriber3. The alert subscriber1 collects alert logs from anomalies detected and forwards them to the IMS. The subscriber2 sends the received alert messages to the namespace orchestrator, which triggers the orchestration process on a given substation namespace. The alert manager subscriber3 sends the received alert messages to the application programming interface (API) of the central management application [18].

### 4.2.6. Mesh-Type Configuration Network

Mesh-type configuration network refers to a network topology in which each device in a network is connected to multiple other devices, forming a mesh-like structure. This type of network is often used in self-healing because it provides redundancy and resilience against network failures. In a mesh-type configuration network, if one device fails, the network can automatically reroute traffic through an alternate path, maintaining connectivity and availability. This approach allows the network to operate seamlessly, even when specific devices or links are unavailable. A proposal by [33] uses a mesh-type configuration to achieve the full potential of a power restoration scheme. The proposal's aim is achieved by deploying the autonomous self-healing technique. However, despite the concerns of the critics of the proposal, who argue that a mesh-type configuration network introduces complications, is costly, and has a higher probability of bi-directional fault current flow,

recent studies have shown that a mesh-type configuration network provides flexibility and security and performs independently. Mesh-type configuration network, by its design, establishes multiple points of failure, making the system resilient and making coupling or decoupling processes easily accessible. Using a mesh-type configuration network ensures the decentralisation of systems management and paves the way for the independent control of multi-level architecture. Mesh-type configuration was demonstrated in [30] implementation of wireless mesh networks (WMNs), which are multi-hop wireless networks with instant deployment, self-healing, self-organisation, and self-configuration features. WMNs offer multiple redundant communication paths throughout the network, and the network automatically reroutes packets through alternative pathways in situations where failure occurs in one section of the network or any threat that interferes with a particular path.

### 4.2.7. Agent Architecture

Agent architecture is the design and implementation of intelligent software agents that can autonomously detect and respond to faults or errors in a system. These agents are designed to operate in a distributed system, with each agent responsible for monitoring a specific aspect of the system and taking action to resolve issues when they arise.

Agent architecture is described in [31] as a rule-based architecture that uses an "if-then" rationalising scheme, in which the consequent result relies on prior experience. Then, appropriate agent architecture influences how well the system's agents handle their operating environment through inference reasoning. The scope for simultaneous internal and external monitoring of the system is limited due to what is referred to as a threading obstacle with JADE/JESS integration, as noted in Cardoso [31]. The alternative and viable option is to structure agent communication from JADE agents to the JESS inference by granting JESS access privileges to the agent communication language (ACL) message or allowing JESS to add ACL message objects to the JESS working memory. Agent architecture in a self-healing system typically consists of the following components:

- Sensing: Agents can monitor the system and gather data about its current state. The data may include system performance metrics, error logs, and other relevant information;
- Diagnosis: Agents use data gathered during the sensing phase to analyse the system's current state and identify any faults or errors. Various techniques can be employed to achieve this, such as comparing current data to historical trends or utilising machine-learning algorithms to detect and pinpoint abnormal behaviours;
- Decision-making: Once a fault or error has been detected, agents must decide how to respond. In such scenarios, the process may involve selecting from a predetermined set of response options, such as restarting a process or diverting traffic to a backup system;
- Action: Agents take action to resolve the fault or error using pre-define or adaptive responses. This process may involve coordinating with other agents to initiate a synchronised response or adjusting the system configuration to prevent future occurrences;
- Learning: Agents continually learn from their experiences and adapt their behaviour over time to improve effectiveness. To adapt effectively, agents may need to adjust their response strategies based on the outcomes of past responses and update their system models using new data.

### 4.2.8. Host Intrusion Detection System on IoT

The host intrusion detection system (HIDS) is an approach for protecting IoT devices against threats. As noted by [37], each device performs this by installing detection agents. The alternative to HIDS is the network detection system (NIDS), which is the most common approach and more scalable because it does not require software installation on the IoT device. However, NIDS has limitations when compared with HIDS, such as the limited capability in its detection functionality, especially in situations where traffic on the IoT network is encrypted. A framework proposed by [37] can detect systems attacks in real-time and react to remediate the effects of such attacks. The framework is an automatic and collaborative host-based self-healing mechanism for IoT devices. The framework

description involves using HIDS to protect a collective instance of IoT devices built into an IoT environment, such as a smart city. In a research study by [37], multiple IoT devices collaborate to train a deep-learning model. The best possible remediation is triggered once the HIDS issues an alert to the model, fulfilling the framework's self-healing functionality. The self-healing architecture proposed consists of three modules: HIDS, health monitoring, and auto-remediation. The HIDS module collects information from the IoT devices, analyses it, and determines if a threat could compromise the IoT device. The health-monitoring module is responsible for assessing the health state of the IoT device by collecting multiple data sources, such as memory usage, disk space, network metrics, etc. The auto-remediation module acts to remedy the effects of malfunction or intrusion of the IoT device.

### 4.2.9. Multi-Area Microgrid

Multi-area microgrid refers to a distributed energy system consisting of multiple interconnected microgrids that can operate independently or in coordination. This type of microgrid is often used to provide energy to various buildings or communities, and it can be designed with self-healing capabilities to improve its resiliency and reliability. A two-area microgrid was proposed by [26] with modes that stand independently. The multi-area microgrid is used to analyse a multi-machine system. A multi-area microgrid was selected to separate the core system into sections, in a fault event simulation, in a manner that allows the application to adopt distributed control. Each area of the multi-area microgrid is equipped with dispatchable and non-dispatchable distributed generators, respectively. The multi-area microgrid implementation separates the system into sections, making asserting control in a system fault event easier. In an experiment, Ref. [26] deployed three diesel power plants and one hydropower plant, all based on synchronous generators. Two types of power load implementation on power plants were deployed: controlled and uncontrolled. The proposed approach utilises machine-learning techniques to detect event signatures in the power system features. A multiclass classification algorithm was then applied to the generated feature data and facilitated self-healing functionality through postfault decision-making that restored the standalone microgrid system without the need for intervention by the central power station.

### 4.3. Machine-Learning Algorithms

Machine-learning (ML) algorithms can be used individually or in combination with each other to create more accurate and comprehensive models of system behaviours. By analysing system data in real-time, ML algorithms can enable self-healing functionality in cyber–physical systems to detect, diagnose, and potentially correct faults or failures before they cause significant damage or downtime. Table 3 lists the state-of-the-art algorithms used in self-healing systems, utilising sensing, mining, and prediction.

**Table 3.** Classification of the existing self-healing algorithms.

| Usage | Algorithms |
|---|---|
| Sensing | • Support Vector Machine (SVM)<br>• Genetic Algorithm<br>• Dynamic Detection Algorithm |
| Mining | • Supervised Knowledge-Based Algorithm<br>• DBSCAN Algorithm |
| Prediction | • Auto-Regressive Moving Average (ARMA)<br>• Long Short-Term Memory (LSTM)<br>• Multi-Layer Perceptron (MLP)<br>• Naïve Bayes |
| Decision | • Monte Carlo Tree Search<br>• Random Forest |

### 4.3.1. Monte Carlo Tree Search

Monte Carlo tree search (MCTS) is an algorithm used in decision-making processes, especially in situations where the outcome of an action is uncertain. It works by constructing a tree of possible future game states and then simulating many random games from those states to determine which actions will most likely lead to successful outcomes.

MCTS can be used to decide how to recover from failures in a complex system. For example, suppose that an extensive computer network experiences a loss in one of its nodes. The self-healing system would need to determine the best course of action to recover from this failure, considering factors such as the system's current state, the possible causes of the loss, and the likely effectiveness of different recovery strategies. MCTS is described by [1] as a method for making optimal decisions when resolving artificial intelligence problems. It combines stochastic simulation and the accuracy of a tree search. The algorithm builds the search tree of the node through a substantial number of random samples, then formulates greater insight into the system and extracts the datasets to calculate the optimal strategy. For example, when faced with unexpected risks in the power grid scenario, MCTS choose the optimal strategy configuration through sampling and estimation results. As the number of samples increases, the obtained strategy will be closer to the optimal approach. MCTS can be used in the context of self-healing by constructing a tree of possible recovery strategies, simulating the effects of each strategy on the system, and then selecting the strategy that is most likely to result in successful recovery. The simulation process can consider factors such as the likelihood of further failures, the time required for each strategy to take effect, and the potential impact on other system parts. MCTS, a powerful tool for self-healing systems, enables the system to make informed decisions in complex and uncertain environments. However, it is essential to note that MCTS is only effective as the quality of the models used to construct the tree may require significant computational resources to run effectively in large-scale systems.

### 4.3.2. Deep Learning

Deep learning (DL) is a subfield of ML that uses neural networks with multiple layers to extract high-level features from raw data. It can be used to develop models that automatically detect and diagnose failures in complex systems and then take appropriate actions to recover from them. One application of DL in self-healing systems is in predictive maintenance, where ML models are trained to detect anomalies in the system data that may indicate potential failure. The models can be trained on historical data to learn the system's expected behaviour and then use that knowledge to detect deviations from normal behaviour that may indicate a failure. Once a failure is detected, the self-healing system can take appropriate actions to prevent or mitigate the effects of the failure. A study by Zhiyuan in [1] demonstrated that DL is an efficient feature extraction method in machine learning. The feature of deep learning aims to establish a deep structure model by integrating the more non-representational feature of data and achieving more detailed characteristics of the data. The three main aspects of DL are unsupervised training, data sample alignment, and data sample testing. The authors of [1] noted that the essence of DL is to find out more valuable features of the dataset by constructing an ML model within many hidden layers and an extensive training dataset to improve the accuracy of classification and prediction. The predictive nature of DL can be deployed to perform systems fault diagnosis, where ML models can be used to identify the root cause of a failure. These models can be trained to analyse sensor data or other system inputs to identify patterns associated with specific types of failures. Once the root cause is identified, the self-healing system can take appropriate actions to address the underlying issue and prevent future failures.

### 4.3.3. Intensive Learning

The purpose of intensive learning, Ref. [1] noted, is to arrive at a perfect to the maximum decision. In the scenario discussed and relating to a power grid, according to the prevailing risks, optimal strategy can be achieved using the Monte Carlo tree search, and

the best solution can be derived from it, producing a new state of the power generating grid. Through comparing the two conditions in an intensive learning process, the evaluation function is updated through the completion of the learning process and repeat iteration, which allows the strategy to meet the expected self-healing functionality of the system. The idea behind intensive learning is to use ML algorithms to learn from large amounts of network data to detect anomalies and predict failures. This approach involves collecting a wide range of data from the network, such as performance metrics, configuration data, and log files, and using these data to train ML models. Once models are trained, they can be used to detect anomalies in the network, such as unusual traffic patterns or unusual changes in configuration settings. The models can also be used to predict when failures are likely to occur, allowing the system to take proactive measures to prevent those failures from happening. Zhough, in [11], suggests that this approach is well-suited for self-healing systems, as it allows the system to learn and adapt to changes in the network environment over time. By continuously collecting and analysing network data, the system can improve its accuracy and effectiveness in detecting and responding to anomalies and failures. The use of intensive learning in developing self-healing systems represents a promising new approach to autonomous network management, which is more data-driven and adaptable than the traditional rule-based systems.

### 4.3.4. Multi-Layer Perceptron (MLP)

Multi-layer perceptron (MLP) is an artificial neural network (ANN) inspired by the structure and function of biological neural networks. MLP consists of interconnected nodes, or "neurons", organised into layers. Data are fed into the input layer and pass through one or more hidden layers before reaching the output layer, where the network makes predictions and has been chosen as one of the models in [17]'s experiment due to its capability of modelling complex nonlinear correlations. Their experiment utilised a single-output MLP with a feed-forward and a fully connected three-layer network. The primitive selector determined the inputs and relevant primitives, and the output corresponded to the quality of service (QoS). MLP can be trained with an arbitrary quality dataset, indicating the potential accuracy of model predictions. It excels in detecting and diagnosing faults or anomalies in a system and can take corrective actions to address them. By continuously training with new data, MLP can adapt to changing conditions and improve its accuracy and effectiveness in detecting and responding to faults. This adaptability makes MLP particularly suitable for self-healing systems. The main limitation of MLP is its computational expense, especially when dealing with large and complex systems. Training MLPs can be resource-intensive, but advancements in hardware and software, such as parallel processing and cloud computing, have made training more efficient and practical for self-healing systems.

### 4.3.5. Supervised Knowledge-Based Algorithm

A supervised knowledge-based algorithm (SKBA) is an ML algorithm that combines expert knowledge with data-driven methods to perform classification or prediction tasks. SKBAs are typically used in situations where the amount of available training data is limited, and expert knowledge can be used to guide the learning process. SKBA is described by [33] as a learning tool that stores the previous fault and restoration data and compiles coincidental solutions. The algorithm is tuned to recognise the cause of fault, develop an imperative plan for restoration and execute restoration operations. The authors of [33] adopted a methodology that uses SKBA to formulate a restoration strategy in an active power grid. SKBAs can be used to detect and diagnose faults in a system and then take action to correct them. The algorithm typically involves a three-step process:

- Knowledge acquisition: Expert knowledge is collected and formalised in a knowledge base, which typically includes rules or heuristics for diagnosing faults;
- Data acquisition: Data are collected from the system, including sensor data, performance metrics, and other relevant information;

- Model training: The SKBA is trained using the knowledge base and the available data. The model is then used to detect and diagnose faults in the system based on the input data.

The execution of SKBA on an automated restoration scheme, as noted in [33], requires a suitable understanding of resources that can be deployed to prevent fault events and deterioration of QoS. Automated restoration strategies are widely deployed in active power grid networks, ensuring the serviceability of electricity and constant power supply to consumers. Implementing self-healing functionality using a knowledge-based algorithm in guided sequential strategies provides the power grid operators with reliable knowledge of the network parameters from MATLAB to be able to visualise the origin of network overloading on the network, according to [33]. The proposed restoration algorithm is deployed with suggested strategies, rerouting proceedings, and distributed generator deployment. The proposed approach creates the possibility of network stability and transmission continuity. The SKBA in [33] is illustrated in (Figure 9) and conceptualised based on the following guidelines:

1. Detect the overloaded transmission lines in the power network;
2. Identify the affected buses that have overloaded transmission lines connected;
3. Identify the busbar with the highest reserve capacity factor to serve as a candidate for the restoration strategy;
4. Identify the nearest distributed generator located near the overloaded transmission line;
5. Identify the overloaded line termination;
6. Establish line connectivity using the highest reserve capacity busbar index.

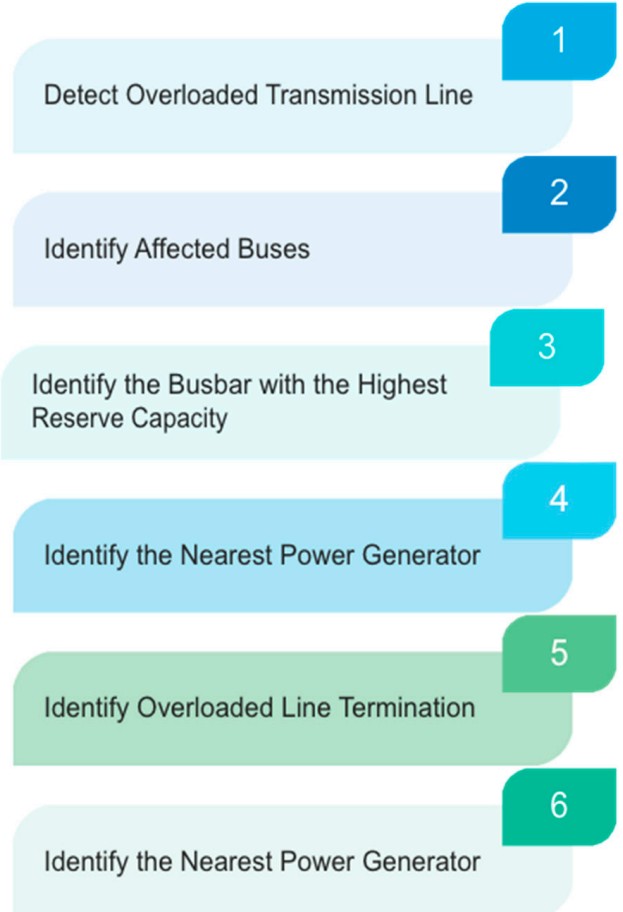

**Figure 9.** Knowledge-based supervised learning for power grid.

### 4.3.6. Artificial Immune System

Artificial immune system (AIS) is a class of computational models inspired by the biological immune system. In AIS, the immune system is modelled to perform specific tasks such as pattern recognition, anomaly detection, and fault diagnosis. AIS can detect and respond to faults or anomalies in a system. The AIS consists of two main components: the detector and the effector. The detector component of the AIS works by comparing the system's current state to a reference model. If the current state deviates from the reference model, the detector identifies it as an anomaly. This process is like how the biological immune system detects foreign agents such as viruses or bacteria. According to Rufus and Esterline, AIS mimics how biological mechanisms fight unknown threats, like how an organism resists a new virus and mimics the process in a computer system [31]. The effector component of the AIS then takes action to correct the anomaly. Depending on the monitored system, the effector can take various forms, including triggering a backup system, adjusting settings, or rerouting traffic. The effector can also learn from previous actions to improve its effectiveness in responding to future anomalies. An advantage of AIS is its ability to adapt and learn from new information, like the biological immune system. AIS can learn from past experiences to improve its accuracy in detecting and responding to faults.

Additionally, AIS can be designed to be fault tolerant, allowing the system to continue to function even if some components fail. However, a potential limitation of AIS is its complexity, which can make it challenging to implement and optimise. Additionally, the effectiveness of AIS depends on the availability of a suitable reference model and the ability to tune the system's parameters.

Existing research in this area of developing AIS proposes several approaches, but the most common are:

- Negative selection: When an anomaly is detected based on the classifying entities being part of the "non-self" originating system;
- Positive selection: When an anomaly is detected based on the classification of the "self" originating system;
- Danger theory: This approach raises the alarm if a harmful signal is detected, regardless of whether the entity is of "self" or of "non-self" of the originating system.

### 4.3.7. Behavioural Modelling Intrusion Detection System

The behavioural modelling intrusion detection system (BMIDS) algorithm analyses familiar simulations within a friendly environment to detect anomaly activities. It is an intrusion detection system that uses ML techniques to monitor system behaviours and detect anomalous activity. BMIDS works by analysing behaviour patterns in the system logs and network traffic and comparing them to expected behaviour based on a model of normal system behaviour. BMIDS can detect and respond to attacks or intrusions in a system. Once an anomaly is detected, the system can mitigate the threat by blocking the offending traffic or alerting security personnel. The functionality is explained in Arrington in [31], where BMIDS is shown to use a computing process to detect intrusion within a smart house environment. A three-stage method for connecting real-world scripted events schemes called behavioural script event scheme (BSES) was presented. BMIDS is a subset of IDS and AIS, which includes in functionality the mechanisms for a subset of IoT intrusions detection systems. BMIDS executes its processes effectively because it creates behavioural models from the BSES dataset, in which use-case IoT devices capture the behaviour of a given anomaly situation, thereby allowing the system to correctly associate the anomaly event with a scripted behaviour and recreate the instance of BSES. The main advantage of BMIDS is its ability to detect previously unknown attacks, also known as "zero-day attacks", that may not be detectable using traditional signature-based approaches. BMIDS can also adapt to changes in system behaviour over time, improving its accuracy and effectiveness in detecting and responding to attacks. However, one limitation of BMDIS is its reliance on accurate models of expected system behaviour. If the model is not comprehensive enough

or the system undergoes significant changes over time, the BMIDS may produce false positives or miss essential threats. Additionally, BMIDS can be computationally expensive to train and maintain and may require significant expertise to configure or optimise. BMIDS can be used with other self-healing techniques, such as automatic system configuration or network rerouting, to help restore the system to a healthy state after an attack.

### 4.3.8. Genetic Algorithm

Genetic algorithm (GA) is an evolutionary algorithm that uses natural selection and genetic operators, such as mutation and crossover, to search for optimal solutions to a problem. GA can be used to optimise system configuration or find the best recovery strategy for fault or failure. GA creates a population of candidate solutions and evaluates their fitness based on a fitness function that measures how well the solution meets the desired criteria. The fitness solutions are then selected to produce offspring through genetic operators, and the process is repeated until a satisfactory solution is found. A genetic algorithm was utilised by [37] to train long short-term memory (LSTM) models in their study. To apply the GA, the set weight matrix that represents the LSTM model of each device was extracted and transformed into a vectorial representation by concatenating all the columns of the weight matrix together into one vector. The devices share their health scores and neural network weights using the blockchain framework so that each device can use it.

For the fitness score, the health score provided by the health-monitoring module was used, and the selection procedure in the GA was performed by selecting just the weight vectors that belonged to the devices with the highest health scores. The combination procedure in the GA was accomplished by taking the weighted means of a small random group from the selected weight vectors. Taking the weighted mean of random groups will ensure that the population of the LSTM model's weight vectors will be diverse and not converge into a single-weight vector. The mutation procedure added random noise drawn from a normal distribution. The GA implementation utilises the same blockchain infrastructure as the CIOTA framework. The utilisation of the same blockchain infrastructure as the CIOTA framework in the GA implementation signifies that the algorithm operates decentralised, eliminating the requirement for a central server for distribution. In simpler terms, the algorithm is executed locally on IoT devices within the network, with each device generating its distinctive model. The IoT devices share their health score and neural network to make the genetic algorithm distributed.GA can be used to improve system resilience and reduce the likelihood of faults or failures, with one advantage being GA's ability to search an ample space of possible solutions and find optimal solutions quickly. GA can also adapt to changes in the system over time, allowing it to optimise system performance continuously. However, a potential limitation of GA is its reliance on a well-defined fitness function, which can be difficult to specify in some situations. GA may also require significant computational resources, especially for complex or large systems.

### 4.3.9. Hybrid Calibration Algorithm

A hybrid calibration algorithm (HCA) is an optimisation algorithm that combines different optimisation techniques to find optimal solutions. HCA can calibrate system parameters or identify the best configuration for a system based on performance criteria. It combines different optimisation algorithms, such as genetic algorithms, particle swarm optimisation, and simulated annealing, into a hybrid approach that exploits the strength of each algorithm. The different algorithms are typically used sequentially or parallel, with the output of one algorithm serving as an input to the following algorithm. HCA was first proposed by [15], combining two direct search algorithms of the Nelder–Mead simplex and Hooke–Jeeves pattern search methods. Nelder–Mead is a popular algorithm for its ability to vary the search directions in the response space at each iteration, while Hooke–Jeeves maintains a well-conditioned search. The hybrid calibration algorithm leverages the advantages of Nelder–Mead and Hooke–Jeeves algorithms to provide robust calibration

performance for the upward dimensional self-healing radio frequency-integrated circuit (RFIC) calibration problems encountered in their experiment. The HCA leverages the advantages of both methods to provide robust calibration performance for the relatively high-dimensional self-healing RFIC calibration problems often overlooked in the real world. HCA can calibrate system parameters, such as tuning the control parameters for a feedback control system, to improve system performance and resilience. HCA can also identify the best configuration for a plan based on performance criteria, such as selecting the best combination of hardware or software settings. An advantage of HCA is its ability to combine different optimisation techniques to find optimal solutions quickly and efficiently. HCA can also adapt to changes in the system over time. The potential limitation is its complexity, making it challenging to implement and optimise. It may also require significant computational resources, especially in complex or large search spaces, coupled with its reliance on well-defined functions, which can be difficult to specify in some situations.

### 4.3.10. Dynamic Event Detection Algorithm

The dynamic event detection algorithm (DEDA) is an algorithm that uses ML techniques to detect and classify events in real-time. DEDA can identify abnormal behaviour or circumstances that may indicate a fault or failure in the system. It works by analysing system data, such as logs or sensor readings, and using an ML algorithm to identify patterns and trends in the data. The algorithm can then classify events as normal or abnormal based on identified patterns. DEDA and a modified ensemble of bagged decision trees with an added boosting mechanism based on a machine-learning algorithm were proposed by [26]. The algorithm interprets the dynamic events and decomposes such events into user-specific field regions to facilitate decision-making in restoring unstable power stations. The novel algorithm, as proposed, can detect patterns in the dynamic data and distinguish the data based on the underlying events. Once the underlying event is detected, the algorithm decides locally on each power generating station and restores the system after a major fault event. The algorithms are independent of each other, as they are installed separately on each power generating station. DEDA can detect events that may indicate fault or failure in the system, such as a sudden increase in CPU usage or a spike in network traffic. Once an abnormal event is detected, the system can mitigate the threat, such as shutting down a component or alerting security personnel. An advantage of DEDA is its ability to adapt to changes in the system over time, allowing it to monitor and detect new events continuously. It can also be used with other self-healing techniques, such as automatic system reconfiguration or network rerouting, to help restore the system to a healthy state after the event. A potential limitation is its reliance on accurate and timely data. DEDA may produce false positives or miss essential events if the data are incomplete or delayed. DEDA can be computationally expensive, especially for large-scale systems or high-volume data streams, and may require significant expertise to configure or optimise.

### 4.3.11. Support Vector Machine

Support vector machine (SVM) is a supervised ML algorithm that can classify and predict system behaviour, identifying potential faults or failures before they occur. This approach can be especially beneficial in systems that demand high availability and reliability [24]. SVM's primary feature lies in its classification and regression analysis ability. It achieves this by identifying the hyperplane in a high-dimensional space that maximises the margin between two classes of data points. SVM can be trained on historical data to identify patterns and predict future behaviour. SVM was described by [39] as a set of supervised prediction-learning methods that are used for classification and regression. The technique uses machine-learning theory to maximise the predicting accuracy of an anomaly on cyber–physical systems. The algorithm supports empirical performance and the structure risk minimisation (SRM) principle. SRM is argued by [39] to be superior to the traditional empirical risk minimisation (ERM) principle. SVM uses statistical-learning

theories to study the problems of knowledge gain, predictions, and decision-making for a given dataset. To build the self-healing functionality whereby virtual machines protect themselves against malware attacks, anomaly patterns of the malware attacks are fed into SVM, a supervised-learning algorithm. Three classifiers, including SVM, RF, and ELM, were applied by [22] to prove that intrusion detection can be achieved and extends the possibility of realising the self-healing functionality goal of their research. SVM is noted to have initially been proposed by Vapnik in [22]. The SVM algorithm is designed to apply to both linear and nonlinear dataset classification tasks. In addition to developing self-healing functionality, SVM has been proven through further research as an effective tool for creating image processing and pattern recognition applications. SVM is a capable tool for creating multiple hyperplanes in high-dimensional space. Then the best hyperplane in the space is then selected to optimally divide data into different classes, with the most significant separation between the classes. SVM finds the optimal line for partitioning class datasets [34–36]. Margins in SVM, Ref. [28] noted, is the distance between the data points of the class. The hyperplane should be chosen, and the margin should be the maximum. The data points on the two classes close to the hyperplane are vectors. The pedicular distance between the hyperplane and the training observations is always calculated, and the shortest of such distances is known as the margin. In other words, the margin is the measurement between the hyperplane and the observation. As such, the maximum margin hyperplane has the highest margin and the most significant distance between itself and the training observation. Testing datasets in SVM are classified using the hyperplane with the most considerable distance.

SVM could be used to monitor the performance of a network and predict when a device is likely to fail. Automated responses triggered by the SVM can be based on various factors, including packet loss, latency, or CPU utilisation. In response to these factors, the SVM can initiate actions such as rerouting traffic to bypass the failing component or restarting the affected service. SVM can also be used to detect and classify security threats. Training SVM on historical data about known security incidents can identify new attacks and predict their likely impact. SVM can serve as a valuable tool for self-healing by enabling the detection of and swift response to issues. These automated responses include blocking traffic from suspicious IP addresses or isolating compromised systems. The result is enhanced self-healing capabilities, allowing for quick and accurate issue detection and response.

### 4.3.12. Naïve Bayes

The naïve Bayes machine-learning algorithm is described by [39] as a classification algorithm based on Bayes theorems. It is also described in Rani in [28] as a classifier that applies the Bayes theorem to solve various classification problems such as detecting systems vulnerability, disease diagnosis, spam prevention, document filtering, etcetera. It is a simple but effective ML algorithm that is particularly useful when dealing with high-dimensional data. The participation features involved in naïve Bayes classification are usually independent as a fundamental principle that guides the algorithm. The naïve Bayes algorithm applies to self-healing systems functionality because it applies to classifying datasets into models that can aid the detection of system vulnerability, providing the first step towards realising a self-healing functionality. The algorithm utilises conditional, and class probability, and the class probability can be the probability of a random dataset belonging to a particular class, which is calculated as:

$$P(C) = \frac{\text{Number of Samples in the Class}}{\text{Total Number of Samples}}$$

The conditional probability, on the other hand, as [39] noted, is the probability of a feature value of a class and is calculated as:

$$P(F|C) = \frac{\text{Number of Frequencies of each Attributes}}{\text{Number of Frequencies of Samples}}$$

All the samples belonging to the classes are compared with each other from the calculated probabilities, and the classes with the most significant probabilities are chosen as the output [39] further argued that the naïve Bayes algorithm performs well on datasets with trivial features because of the probability of the trivial features contributing less to the output. It performs well when making predictions, as it involves only probabilities of features and classes.

Naïve Bayes assumes that each feature is independent of all other features. The algorithm can effectively scale to large datasets by simplifying the computation of probabilities. It can be trained on historical data to identify patterns and accurately predict future behaviour. For example, a naïve Bayes classifier could be used to predict the likelihood of a particular system component failing based on the factors such as CPU usage, memory usage, network traffic, and disk I/O. The classifier could be trained on historical data to learn the relationship between these variables and their impact on the system performance. Once a classifier has been trained, it can diagnose faults or failures in real time. For example, if CPU usage suddenly spikes, the naïve Bayes classifier could predict that a particular component will soon fail. In response to anomalies, this can trigger an automated response, such as workload migration to another server or service restart, facilitating effective self-healing. Naïve Bayes is a valuable tool for self-healing systems, enabling them to detect and respond to faults or failures quickly and accurately. However, it is essential to note that the "naïve" assumption of independence may not always hold in practice; more sophisticated models may be required for complex systems.

### 4.3.13. Random Forest

Random forest (RF) is an ensemble-learning algorithm used in self-healing systems to classify and diagnose system faults or failures. It is a robust ML algorithm that combines multiple decision trees to achieve better accuracy and generalisation. RF is described by [39] as a set of decision trees that tend to produce accurate decisions based on growing multiple trees of different subsets of a dataset. RF is an ensemble classifier used for the regression analysis of intrusion detection in datasets [22,24]. It functions by creating various decision trees in the training phase of the classification process, then produces an output of class labels with a majority vote. The decision trees are independent of each other, but they operate in an ensemble manner, as noted by Sharma [28]. Each decision tree produces an output and a class, and the majority class amongst them becomes the random forest. The RF classifier algorithm applies to self-healing functionality because it can detect the anomaly in the system's critical datasets and trigger its self-healing functions. RF relies on the need to remove trivial features because they do not affect the outcome of the results. The information gain of RF split on a dataset is calculated as:

$$G(T, X) = \text{entropy}(X) - \text{entropy}(T, X)$$

The above equation is explained in [39] as a calculation where "G" is the gain, and the attribute with the highest gain value is chosen as the split/decision node. If the value of the entropy of the selected node is 0, then it becomes the leaf node. The algorithm runs recursively until no more splits are present on the nodes, whereby if the entropy value of the selected node is 0, it becomes the leaf node. RF could be used to predict the likelihood of a particular system component failing based on multiple features such as CPU usage, memory usage, network traffic, and disk I/O. It is a critical self-healing tool, and its ability to handle large datasets and high-dimensional feature space makes it useful for implementing complex self-healing systems.

### 4.3.14. DBSCAN

Density-based spatial clustering of application with noise (DBSCAN) is described by [10] as an algorithm used in detecting anomaly events. An unsupervised clustering algorithm assigns data points to clusters based on their spatial density. The algorithm defines a neighbourhood around each data point and grouping points based on their proximity or density. The purpose of the algorithm is to detect distinct anomalies that belong to the same underlying event based on the anomaly level time-series value of the profile features. For example, it could cluster together data points representing normal system behaviour while leaving outlying data points as noise. The resulting clusters can predict future system behaviour and diagnose faults or failures. The self-healing functionality is achieved using each profiled feature or KPI (key performance indicators) to detect anomalies in timespan using DBSCAN. Observation for a time t is considered abnormal if the anomalous value density cap goes above a given threshold. DBSACN can also identify anomalies in network traffic or security data. By clustering together network traffic data, it could identify unusual patterns of activity that could indicate a potential security breach, which could trigger an automated remedial response.

### 5. Analytical Comparison of MLP, SVM, and RF in Classifying Error in Simulated CPSs

These machine-learning approaches, namely multi-layer perceptron (MLP), support vector machines (SVMs), and random forest (RF), have been identified as the best performing through experimental evaluation. They offer valuable capabilities for self-healing systems by analysing data, detecting anomalies, and making informed decisions for problem resolution [24]. In the context of classifying errors in simulated cyber–physical systems (CPSs), multi-layer perceptron (MLP), support vector machines (SVMs), and random forest (RF) have emerged as main machine-learning approaches. These algorithms have demonstrated superior performance through rigorous experimental evaluations compared to other methods. As a result, they offer valuable capabilities for self-healing systems, where their ability to analyse data, detect anomalies, and make informed decisions becomes crucial for practical problem resolution [24]. MLP, SVM, and RF offer valuable contributions in the context of self-healing systems. By analysing the data collected from the CPS, these algorithms can identify anomalies or errors, allowing for early detection of system failures or deviations from normal behaviour. This proactive approach helps mitigate potential risks and enables timely problem-solving interventions.

Multi-layer perceptron (MLP) accuracy is shown in Figure 10. The model is an artificial neural network (ANN) that plays a crucial role in self-healing applications. In self-healing systems, MLPs analyse various data sources, such as system logs or performance metrics, to detect anomalies and make informed decisions to resolve issues automatically. The MLP model consists of multiple layers of interconnected nodes (neurons). Each neuron applies a weighted sum of inputs, followed by an activation function. The output of the MLP is obtained by propagating the inputs through the network. The overall prediction of the MLP is obtained by combining the results of all neurons in the final layer. The equation for a neuron's output is:

$$output = activation\_function\left(weighted_{sum(inputs)}\right)$$

Figure 11 illustrates the support vector machine's (SVM) accuracy chart. The SVM is a supervised-learning model commonly employed for data classification. In intrusion detection systems [24], SVMs significantly distinguish between normal and malicious network behaviours. This capability contributes to the self-healing approach by facilitating proactive security measures. The SVM algorithm seeks to identify a hyperplane that effectively separates the data into distinct classes. This hyperplane is determined by a

subset of training samples known as support vectors. The sign of the decision function determines the predicted class. The equation for the decision function of SVM is:

$$decision\_function = sign(dot\_product(weights, inputs) + bias)$$

Figure 12 displays the accuracy chart of the random forest algorithm. Random forest is an ensemble-learning technique that leverages multiple decision trees to generate predictions [32]. This algorithm is renowned for its resilience and effectiveness in handling extensive datasets. In the context of self-healing systems, random forests can analyse diverse data sources and detect patterns indicative of system failures or anomalies. Each decision tree within the random forest is trained on a random subset of the data and features. The final random forest prediction is derived by aggregating the predictions of individual trees [32]. The prediction equation in a random forest combines the predictions made by all the trees.

$$prediction = majority\_vote(prediction_{tree_1}, prediction_{tree_2}, \dots, prediction_{tree_n})$$

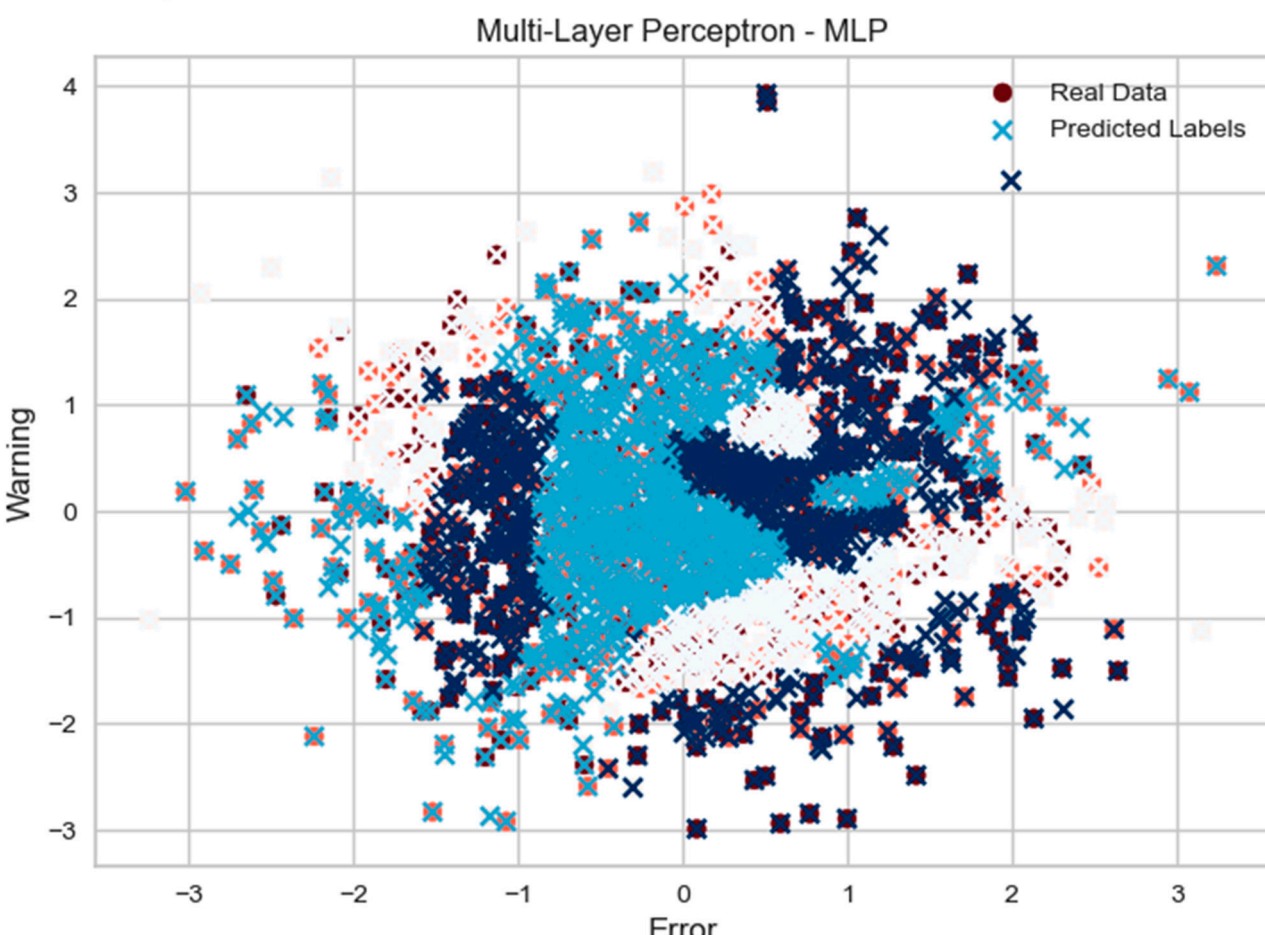

**Figure 10.** Multi-layer perceptron accuracy chart.

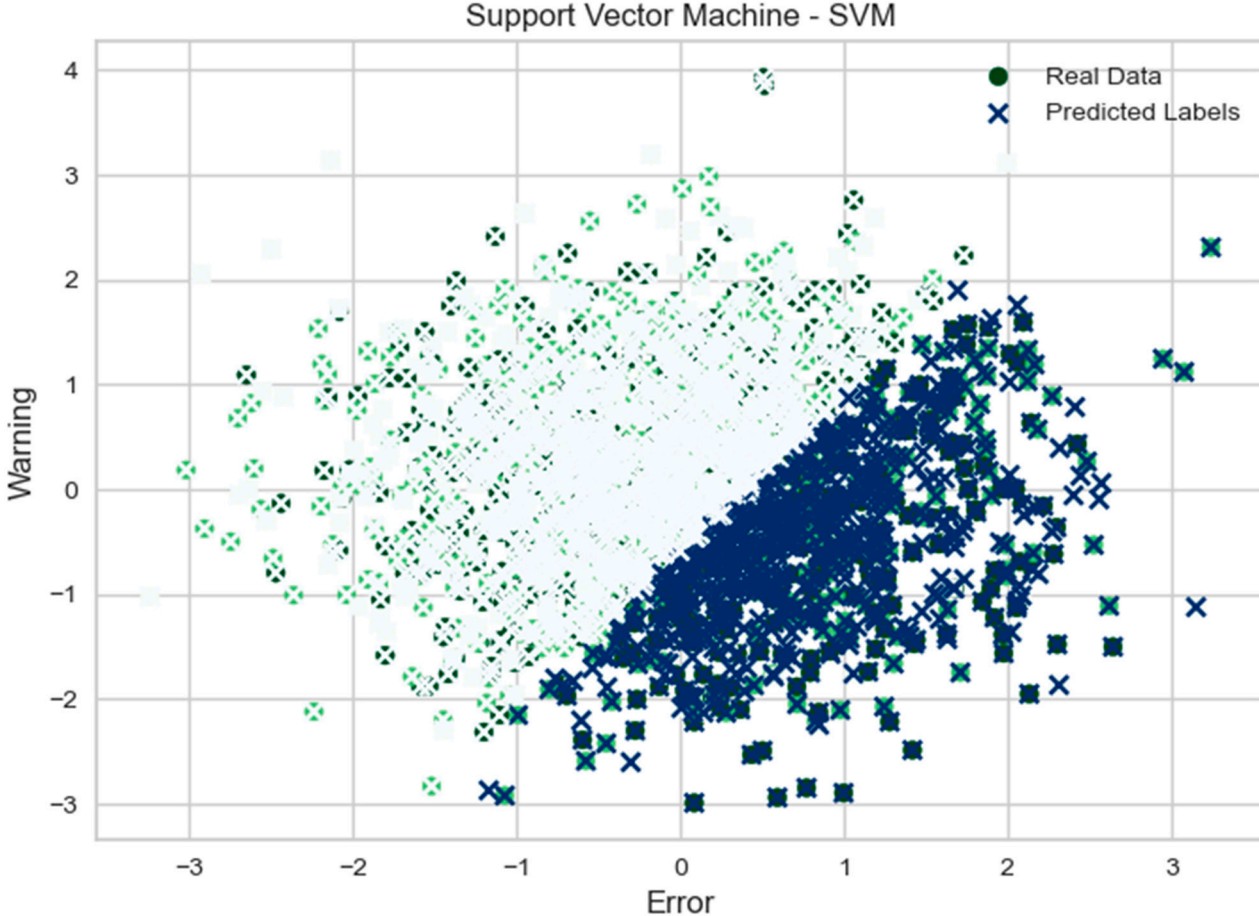

**Figure 11.** Support vector machine accuracy chart.

*5.1. Breakdown of What the Models Chart Plots Represent*

The scatter points in a specific colour (e.g., red, green, or blue) represent the actual data points from the dataset. Each point's position on the chart is determined by its "Error" value on the *x*-axis and its "Warning" value on the *y*-axis. This visualisation helps you see the distribution and patterns in the original dataset. Predicted labels: the scatter points marked with an "x" symbol and a different colour (e.g., blue) represent the predicted labels for the corresponding data points. These labels are obtained by applying the trained model to the input data. The position of each predicted label point on the chart is determined by the same "Error" and "Warning" values as the actual data points. By comparing the positions of the fundamental data points and the predicted label points, you can visually assess how well the model performs in classifying the data. The model makes accurate predictions if the predicted labels align closely with the actual data points.

On the other hand, if the predicted labels are scattered or do not align well with the actual data, the model might not perform well in classification. The specific colour mappings used in the code (cmap = "Reds", cmap = "Greens", cmap = "Blues")indicate different colour gradients that help distinguish the other classes or labels in the dataset. The colour intensity can provide additional insights into the distribution and separation of the data points.

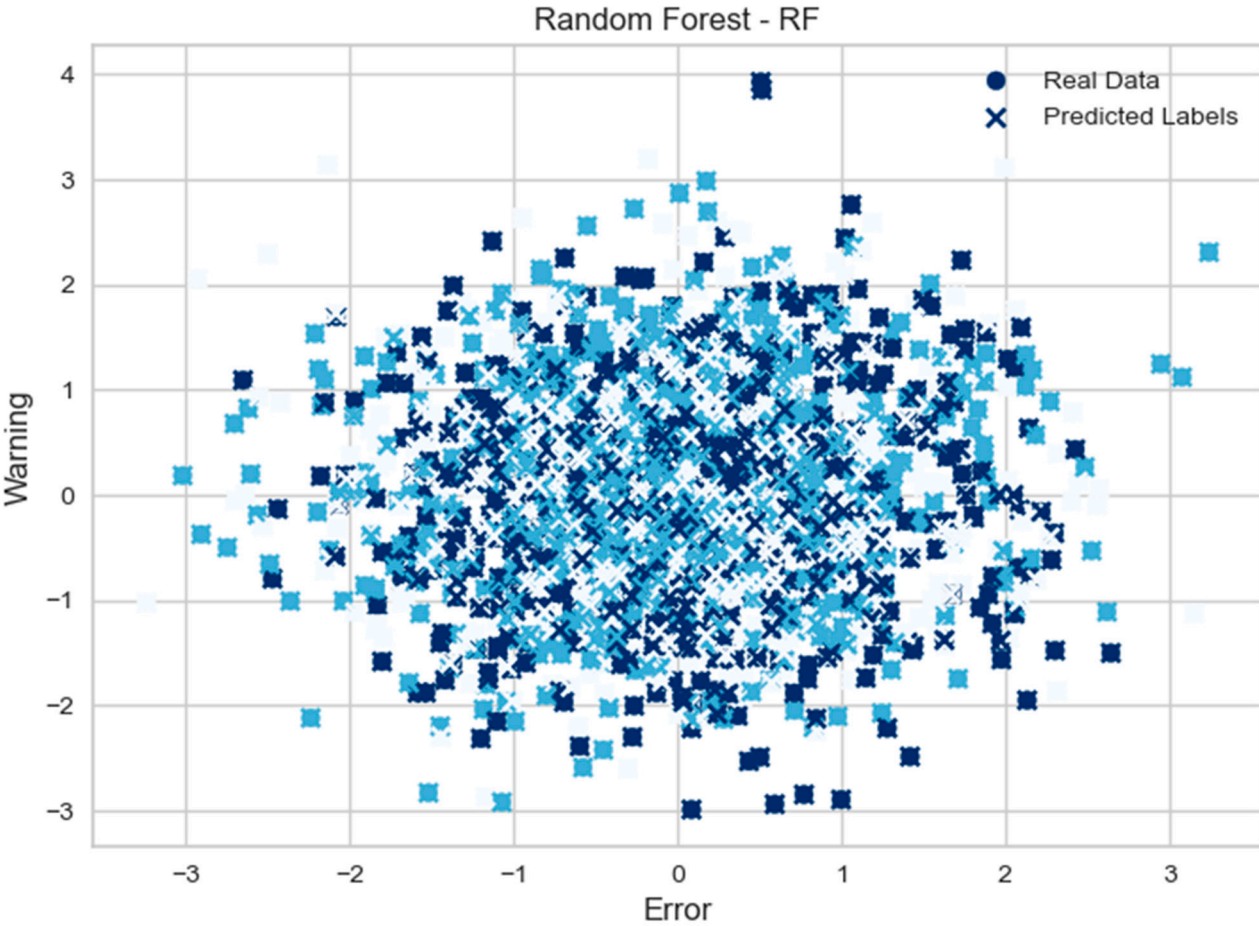

**Figure 12.** Random forest accuracy chart.

*5.2. Experiment Method*

The provided code demonstrates an experiment comparing the performance of three machine-learning models (multi-layer perceptron (MLP), support vector machine (SVM), and random forest) on a classification task using a given dataset. The experiment compares models using the provided dataset, assesses their accuracy, and visualises the results. It also employs the PyCaret library for further model evaluation and selection. The experiment includes the following steps:

1.  Sample Data:

    1.1   A random seed for reproducibility was set.
    1.2   The number of samples as "num_samples" is defined.
    1.3   Random data were generated using "np.random.randn" with dimensions "num_samples" by 2.
    1.4   Random labels ranging from 0 to 2 using "np.random.randint" for "num_samples" times were generated.

2.  Saving the Data to a CSV File:

    2.1   A pandas DataFrame called "df" with columns named "Error", "Warning", and "Label" was created.
    2.2   The DataFrame was saved to a CSV file and specified by "csv_file" using the "to_csv" function.

3.  Loading the Data from the CSV File:

    3.1   The CSV file was read into a pandas DataFrame called "df_loaded".

3.2　The "Error" and "Warning" columns were extracted from "df_loaded" and assigned to "data_loaded".

3.3　The "Label" column was extracted from "df_loaded" and set to "labels_loaded".

4.　Multi-Layer Perceptron (MLP) Example:

4.1　The function "mlp_self_healing" was defined to train an MLPClassifier model on the "data" and "labels".

4.2　Prediction using the trained MLP model and calculating the accuracy was performed.

4.3　The "real" data were randomly generated, and predicted labels were plotted into a scatter chart.

5.　Support Vector Machine (SVM) Example:

5.1　The function "svm_self_healing" was defined to train an SVC model with a linear kernel on the "data" and "labels".

5.2　Prediction using the trained SVM model and calculating the accuracy was performed.

5.3　The "real" data were randomly generated, and predicted labels were plotted into a scatter chart.

6.　Random Forest Example:

6.1　The function "random_forest_self_healing" was defined to train a RandomForestClassifier model with 100 estimators on the "data" and "labels".

6.2　Perform prediction using the trained random forest model and calculate the accuracy was performed.

6.3　The "real" data were randomly generated, and predicted labels were plotted into a scatter chart.

7.　Calling the Self-Healing Approaches Using the Loaded Data:

7.1　The "mlp_self_healing" function with "data_loaded" and "labels_loaded" was invoked.

7.2　The "svm_self_healing" function with "data_loaded" and "labels_loaded" was invoked.

7.3　The "random_forest_self_healing" function with "data_loaded" and "labels_loaded" was invoked.

8.　Preparing the Data for PyCaret:

8.1　Pandas DataFrame called "df_pycaret" was created by concatenating "data_loaded" and "labels_loaded" along the columns.

8.2　The column names ("Error", "Warning", "Label") were set.

9.　Initialising PyCaret Classification Setup:

9.1　PyCaret "setup" function to initialise the classification task with "df_pycaret" as the dataset and "Label" as the target variable was used.

10.　Comparing Models and Select the Best One:

10.1　PyCaret's "compare_models" function was used to compare the performance of the available models (MLP, SVM, RF).

10.2　The best-performing model based on the comparison was selected.

11.　Evaluation of the Performance of the Best Model:

11.1　PyCaret's "evaluate_model" function was used to evaluate the performance of the best model selected in the previous step.

The experimental results of the PyCarat analysis for different models are shown in Table 4. PyCarat is a performance analysis tool that provides insights into model performance using various evaluation metrics. The figure presents the comparative analysis of different models based on their performance measures such as accuracy, AUC, recall, precision, F1 score, Kappa, MCC, and time taken for predictions. It helps assess and compare the models' effectiveness in the given context. The table represents the performance metrics of different models. Here's an explanation of the metrics:

- Model: The name or identifier of the model.
- Accuracy: The proportion of correctly classified instances by the model.
- AUC: The Area Under the Receiver Operating Characteristic (ROC) curve measures the model's ability to distinguish between classes.
- Recall: Also known as sensitivity or true positive rate, it represents the proportion of true positive predictions out of all actual positive instances.
- Prec.: Short for precision, it indicates the proportion of true positive predictions out of all predicted positive instances.
- F1: The harmonic mean of precision and recall provides a balanced model performance measure.
- Kappa: Cohen's kappa coefficient assesses the agreement between the model's predictions and the actual classes, considering the agreement by chance.
- MCC: Matthews Correlation Coefficient, a measure of the quality of binary classifications.
- TT (Sec): The time taken by the model to make predictions (in seconds).

**Table 4.** Models PyCarat analysis experiment result.

|  | Model | Accuracy | AUC | Recall | Prec. | F1 | Kappa | Mcc | TT (S) |
|---|---|---|---|---|---|---|---|---|---|
| mlp | MLP classifier | 0.3493 | 0.5215 | 0.3493 | 0.3498 | 0.3327 | 0.0201 | 0.0210 | 0.1620 |
| svm | SVM-Linear Kernel | 0.3357 | 0.0000 | 0.3357 | 0.3007 | 0.2735 | 0.0052 | 0.0047 | 0.0670 |
| rf | Random Forest Classifier | 0.3336 | 0.4965 | 0.3336 | 0.3349 | 0.3334 | −0.0002 | −0.0002 | 0.1950 |

Based on the table, it can be observed that the MLP Classifier (mlp) has the highest accuracy (0.3493) and AUC (0.5215), while the Random Forest Classifier (rf) has the highest MCC (0.1950). The SVM with a linear kernel (svm) performs less across multiple metrics.

## 6. Discussion

For the self-healing of cyber–physical systems to be effective, resilience must be at its core, just as defined in [25], where it is noted to include system monitoring, adaptation, redundancy, decoupling, and focus at the edges but simple at its core. ML is a notable toolset widely accepted in various research as a critical aspect of implementing self-healing functionality in computer systems. For example, as recent research has shown when combined with a fault-solving strategy network, an ML algorithm can o implement self-healing functionality in cyber–physical systems. The primary language for implementing ML algorithms is the R language, as presented by Bodrog in [25], and libraries such as Keras or Tensorflow are readily available to transform R language into other languages such as Python or C++. To overcome the problem of real-time systems self-healing and attack remediation issues, a framework for automated remediation triggers is possible, just as proposed by [37], and it details how self-healing can be built into the Internet of Things (IoT) devices. IoT has become popular in homes, offices, and schools, making it very important to construct self-healing devices to provide effective and uninterrupted services. The first factor to consider when developing a self-healing cyber–physical system is detecting anomalies and fault events. Naïve Bayes, artificial neural networks, convolutional neural networks, deep learning, etc., are some of the algorithms available to researchers to utilise in training anomaly datasets and to create a classified pattern of threat events. The dilemma in making this consideration is how to prevent the algorithm chosen from flagging up business-as-usual events wrongly as anomaly events; hence, the ML algorithm must be able to differentiate between these events. Danger theory is one approach that can be deployed to resolve the false-negative issues, in the sense that a threat alarm can only be raised when there exists the presence of potentially harmful events that the system is not familiar with and thereby eliminate "zero-day" attacks or in other words, attacks that the system had not seen previously.

The next step after detecting anomaly events when implementing a self-healing cyber–physical system is to be able to trigger warnings to other parts of systems and or other

processes within the system components or the network. An alert trigger can be activated similarly as described in [18] using alert management that relays system status to the intrusion mitigation system and then takes actions to stop the presence of potential threats through a programmed remediation process. In this instance, intrusion mitigation encompasses removing the identified threat event and restoring the system to its stable state. The system's data module profiles the normal state of the system and creates datasets for ML training derived from the system logs. There are options for creating a dictionary of possible attack vectors, which the ML algorithm that is primed for detecting anomalies can rely on to ascertain if events fall within a suspicious category class of threats. This approach is the most widely available and studied, as described by [33] in a proposal that uses a knowledge-based algorithm to identify attack vectors. The approach, though, has increasingly become problematic as cyber–physical systems attacks have become more sophisticated. For example, state actors and multinational corporations are increasingly involved in cyber–physical systems breaches. The increasingly sophisticated nature and scope of threats make it challenging to predict or create a practical threat dictionary that covers all possible threats. A more robust approach that is a viable alternative to using an attack dictionary or relying on previously seen threat events to safeguard against new unseen threats is the proposal closely related to it by [37], in which the evolutionary algorithm principle which mimics the natural evolution process where multiple solutions are presented and the most effective selected to advanced and likely to be picked by the algorithm to protect the system based on past effectiveness.

Self-healing functionality is currently being deployed in many countries to automate power infrastructures, such as in the Netherlands, France, Vietnam, and Cuba, to shore up the resilience of their power generation networks. In particular, the FLISR platform for achieving this functionality was proposed by [20] and deployed in these countries. Resilience refers to the ability of a system to withstand and recover from disruptive events, such as cyber-attacks or physical failures, and to continue functioning at an acceptable level. Silvia [23] contributed to the discussion on resilience in cyber–physical systems, mainly focusing on the power grid. Their work sheds light on the challenges and strategies involved in ensuring the power grid's resilience. They emphasise that the power grid plays a critical role in modern society, and its disruption can have far-reaching consequences, affecting not only the provision of electricity but also numerous sectors that rely on its stability. However, the increasing integration of information and communication technologies in power systems introduces new vulnerabilities and potential points of failure.

Therefore, building resilient power grids that can withstand and rapidly recover from disruptions is paramount. The authors outline several critical aspects related to resilience in the power grid. First, they highlight the importance of system monitoring and situational awareness. Operators can detect anomalies or potential threats by continuously monitoring the grid's performance and responding proactively. To accomplish this, advanced sensing technologies, data analytics, and real-time monitoring tools are utilised to gather and analyse relevant information regarding the grid's condition. Another crucial aspect discussed by [23] is the need for practical risk assessment and management. Understanding the vulnerabilities and potential impacts of different disruptions allows for developing appropriate risk mitigation strategies. This process entails identifying critical components of the power grid, analysing their dependencies, and implementing measures to enhance their resilience. These measures can include redundancy, alternative routing, and diversification of energy sources to minimise the impact of failures. The role of advanced technologies in enhancing resilience was discussed in [23]. The discussion highlights the potential of artificial intelligence, machine learning, and blockchain technologies to strengthen the power grid's resilience. These technologies can facilitate rapid decision-making, improve system automation, enhance security, and enable efficient energy management.

Proactive rolling-horizon-based scheduling of hydrogen systems for resilient power grids is a method that focuses on enhancing the resilience of power grids by integrating hydrogen energy systems [40]. Resilience is crucial to power grids, ensuring they can

withstand and recover from natural disasters, cyber-attacks, or equipment failures. The proactive rolling-horizon-based scheduling approach involves making real-time decisions for scheduling the operation of hydrogen systems within a power grid. Hydrogen systems, including electrolysers, hydrogen storage tanks, and fuel cells, play a vital role in integrating renewable energy sources, optimising power generation, and providing backup power during emergencies. This method employs a rolling-horizon-based approach, meaning that the scheduling decisions are made in short intervals, often referred to as time steps, and updated periodically as new information becomes available. This dynamic approach allows flexibility in adapting to changing conditions, which is essential for maintaining resilience in power grids. When combined with [40]'s proactive rolling-horizon-based scheduling of hydrogen systems for resilient power grids, machine-learning techniques can further strengthen the overall resilience of the grid infrastructure. Integrating machine learning with the scheduling method allows for enhanced self-healing capabilities in power grids. Machine-learning algorithms can analyse real-time data from sensors and smart metres to detect anomalies or potential faults in the grid.

These algorithms can identify abnormal behaviour and trigger proactive actions by continuously monitoring grid parameters, such as voltage levels, frequency deviations, or load variations. The machine-learning models, trained using historical data, can predict potential faults or disturbances in the power grid before they occur. The models can generate early warnings or alerts for system degradation or vulnerability by analysing patterns and trends. The scheduling algorithm can then use this information to initiate appropriate responses, such as reconfiguring the grid or activating backup systems, to mitigate the impact of potential disruptions.

Resilience methods are vital in dealing with uncertainties due to the failure of cyber–physical systems. Ref. [41] argues that a proper modelling tool must be used to develop such methods. The modelling tool described by Murata in [41] should be able to capture the characteristics of asynchronous, synchronous, and current events in cyber–physical systems. Petri nets are another class of modelling tool for this purpose, of which many variants have been proposed over the past decades and are widely used in developing different system applications. A nominal model of a class of cyber–physical system and an uncertainty model to test the system's robustness and resilience was proposed [14]. For the nominal model, Ref. [31] used discrete timed Petri nets as the cyber world models to describe the production process of a class of the cyber–physical system with different types of tasks and sets of distinct types of resources. Constructing the cyber world model of the cyber–physical system was easily performed using a bottom-up approach. The bottom-up approach starts with creating the cyber world model of each resource type and task subnet. Discrete timed Petri nets describe each resource and task subnet. Each task subnet describes the production process workflow, and each resource subnet describes the activities or operations the specific type of resources could perform. Self-healing functionality is increasingly improving the critical infrastructure resilience of countries at this very moment despite it being a very recent technological development. The research direction in this area indicates further acceleration of the functionality in private areas, especially in deploying the 5G networks. The cyber–physical self-healing technology no doubt has the potential to revolutionise systems security. All of which would not have been possible without the opportunities presented by machine-learning algorithms. ML algorithms are advancing speedily because of the collaborative nature of modern software development through the wide industry acceptance and adoption of open-source libraries and packages. This phenomenon aids the seamless translation of the core ML language to other languages that provide a variety of choices for developers. Open-source libraries such as TensorFlow, Keras, PyTorch, and OpenCV are some examples of the collaborative efforts of developers in the ML open-source space.

## 7. Conclusions

This paper has reviewed existing self-healing theories and methods, highlighting machine learning as a promising tool for implementing self-healing functionality in cyber–physical systems. Self-healing has significant potential in computing, offering improved uptime, reliability, and performance in critical systems. By automating fault detection and response, self-healing reduces downtime, minimises manual interventions, and enhances the safety and efficiency of critical infrastructure. Self-healing mechanisms can be used in any system where uptime, reliability, and performance are crucial. However, there are still gaps that future research needs to address to accelerate the real-world adoption of self-healing technology. These gaps include expanding the classification of attacks in machine-learning models to improve their effectiveness and exploring the transferability of trained automatic remediation models between multiple IoT devices.

This research has identified several exciting directions and issues for future studies in self-healing technology. These include expanding the classification of attacks in machine-learning models to improve their effectiveness and exploring the transferability of trained automatic remediation models between multiple IoT devices. Adopting IoT, 5G networks, collaborative software development, and knowledge transfers within the field will advance self-healing theories. As novel technologies emerge, self-healing functionality implemented using machine-learning algorithms will enhance cyber–physical systems' security, reliability, and intuitive nature, enabling self-organisation and self-restoration. This paper anticipates a future where self-healing functionality, implemented using machine-learning algorithms, will exponentially make cyber–physical systems more secure, reliable, and intuitive, leading to self-organisation and self-restoration. While there are gaps to be addressed for real-world adoption, the prospects for self-healing technology are promising, and further research and innovation in this area will continue to propel its development and practical application.

**Author Contributions:** Conceptualisation, O.J.; formal analysis, O.J.; investigation, O.J.; methodology, H.A.-K. and A.S.S.; resources, M.A.; supervision, H.A.-K. and A.S.S.; validation, O.K., H.A.-K. and A.S.S.; writing, O.J., H.A.-K. and A.S.S.; review and editing, M.A.T., O.K. and F.A.-O. All authors have read and agreed to the published version of the manuscript.

**Funding:** This work was supported by the Deanship of Scientific Research (DSR), King Khalid University, Abha, under grant no. (RGP.1/380/43). The authors, therefore, gratefully acknowledge the DSR's technical and financial support. Also, we would like to thank the support from the Zayed University fund, grant number R21092.

**Data Availability Statement:** Not applicable.

**Acknowledgments:** The authors would like to thank Nottingham Trent University, King Khalid University, and Zayed University for their support.

**Conflicts of Interest:** The authors declare no conflict of interest.

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
