# Peer review of "Self-Healing in Cyber–Physical Systems Using Machine Learning: A Critical Analysis of Theories and Tools"

_futureinternet, doi:10.3390/fi15070244_

Round 1

Reviewer 1 Report

The submitted manuscript provides a critical analysis of theories and tools of self-healing cyber-physical systems using machine learning.

Introduction:

The section must be more supported by appropriate references.

Figure 1 is a frequently appearing picture of the revolutions in the industry. Has this one been prepared by the authors? If not, please give its source. By the way, its scale must be adjusted.

Section 2:

Figure 2 and 3 is of low quality.

Section 3:

Line 326 – why starting with point 5?

The same problem is in lines 355, 399 and 415.

Figure 5 and 6 – much too big and of low quality.

Section 4:

The approaches must be supported by appropriate references.

Different styles of punctuation are annoying.

Table 1 – what is its purpose?

What is “Vluation Network”?

In the row with “Genetic Algorithm” there is a text coming over it.

Line 668 – improper numbering.

Line 699 – the equation is not clear.

Line 773 – improper numbering.

Fig. 8 – low quality and too big.

Line 945 – improper numbering.

Fig. 9 – low quality and too big.

Line 1253 – improper numbering.

General:

A methodology of research is missing. If aiming at a review paper, it must be clearly stated how the described approaches have been selected. Please add the appropriate description to the manuscript.

References:

The references in the manuscript body are not according to the template.

The style of writing could be improved.

Author Response

Please refer to the attached response letter.

Reviewer 2 Report

The authors propose an extended survey regarding the problematic of self-healing relative to cyber-physical systems. Certain problematic aspects should be addressed.

1. The authors should create a section, or sub-section(for example, in section 5), which selects and analytically compares and analyzes three-four of the most promising reviewed approaches. An article of this length, should help readers understand the scientific problematic as precisely as possible, and help them make an informed decision for their own research and practical purposes.

2. Consequently, the assertions/suggestions made in this new section should be backed by experimental data, which are generated either by the authors themselves through proper simulations, or extracted from the surveyed papers themselves.

3. The list of references, which currently includes 28 entries, is rather undersized for such a review paper. I should be extended with more relevant up to date papers, preferably at leas 10-15 additional papers.

4. The English language should be fully proofread and improved.

The English language should be fully proofread and improved.

Author Response

(The authors gave the same response as above.)

Reviewer 3 Report

The research issues discussed in this paper are interesting and important.

These research issues may attract the attention of relevant community.

This paper is largely presented clearly and well-organized.

I found errors that need to be corrected.

Several missing papers are suggested to be included.

Please refer to my review as follows. 

(1) Line 93-94: There is an error in the following sentence:

To control the costs as networks, migrate to 5G, there is a shift of focus towards automating the systems protection process through the implementation of self-healing.

(2)The first sentence in the paragraph in Line 126-133 as follows should be revised as the four points mentioned in the paragraphs include goal and objective of this paper.

"The contribution of this paper is principally focused on:

 This paper aims to enhance knowledge by highlighting current trends in the area of study.

 The main objective of this paper is to identify the latest machine-learning tools, methods, and algorithms for 

integrating self-healing functionality into cyber-physical systems. 

 The self-healing capability of cyber-physical systems will be evaluated concerning state-of-the-art techniques,

and the use of machine learning tools and methods in implementing self-healing functions will be explored.

 The existing literature will be critically reviewed to identify current tools, methods, algorithms, classification

models, frameworks, networks, and architectures that are currently deployed for a self-healing approach."

(3) This paper is a survey paper. It reviews existing literature and also several future research directions/issues are dissussed.

For example, in Line 496-501 as follows, several future research directions/issues are dissussed.

"Future studies could see the automatic remediation algorithms built into IoT devices that would be capable of providing self-healing system safeguards and reducing the cost of maintenance which can ordinarily occur from system failure or attack. Future work to further explore the theory would be centred around the addition of more classification of attacks to the ML models; to bolster the learning capacity of the models, as well as explore the feasibility of transferring a trained automatic remediation model between multiple IoT devices."

However, these interesting research directions/issues are not summarized in the conclusion section.

I suggest to summarize these interesting research directions/issues are not summarized in the conclusion section.

(4) Line 1248-1250: There is an error in the following sentence:

Open-source libraries such as TensorFlow, Keras, PyTorch, OpenCV etcetera as some examples of the collaborative efforts of developers in the ML

open-source space.

(5) A review paper should broadly include papers relevant to the research subject. 

There are several missing papers on resilience in the context of cyber physical systems/power grids. For example,  resilience in the context of cyber physical systems was discussed in [i] below. Security is only one aspect of resilience in  cyber physical systems. Other properties such as methods to deal with the impacts due to failures in cyber physical systems are also relevant. For example,  these issues are addressed in papers [ii] and [iii] below. 

I suggest to include the following papers and cite them in the main text properly. 

[i] Silvia Colabianchi, Francesco Costantino, Giulio Di Gravio, Fabio Nonino, Riccardo Patriarca,

Discussing resilience in the context of cyber physical systems, Computers & Industrial Engineering,

2021,160, 107534.

[ii] Hsieh, F.-S. An Efficient Method to Assess Resilience and Robustness Properties of a Class of Cyber-Physical Production Systems. Symmetry 2022, 14, 2327. 

[iii] Hamed Haggi, Wei Sun, James M. Fenton, Paul Brooker, "Proactive Rolling-Horizon-Based Scheduling of Hydrogen Systems for Resilient Power Grids", IEEE Transactions on Industry Applications, vol.58, no.2, pp.1737-1746, 2022.

Please refer to my review.

Author Response

(The authors gave the same response as above.)

Round 2

Reviewer 1 Report

The article has been revised according to (most of) my comments. Some issues are, however, still valid:

The numbering of references in the text body should start with [1] and not [31].

The formatting is not consistent in the article.

The figures are still much too big.

The tables are also too big.

The equation in line 565 does not look good.

The numbering starting in line 745 is not correct.

Style of writing could be improved.

Author Response

(The authors gave the same response as above.)

Reviewer 2 Report

The authors marked all the text in the revised manuscript, so it was difficult to discern their changes, as compared to the previous review version. Nevertheless, I appreciate that enough changes were implemented, and the manuscript may be considered for further processing. The English language should undergo another round of proofreading.

The English language should undergo another round of proofreading.

Author Response

(The authors gave the same response as above.)

Reviewer 3 Report

The authors had responded to my previous review and revised the paper thoroughly.

The paper quality has been improved significantly.

Author Response

(The authors gave the same response as above.)
